# Theoretical Learning Performance of Graph Networks: the Impact of Jumping Connections and Layer-wise Sparsification

**Jiawei Sun**                                                                    *sunj11@rpi.edu*
*Department of Electrical, Computer, and Systems Engineering*
*Rensselaer Polytechnic Institute*

**Hongkang Li**                                                                *lihk@seas.upenn.edu*
*Department of Electrical and Systems Engineering*
*University of Pennsylvania*

**Meng Wang**                                                                  *wangm7@rpi.edu*
*Department of Electrical, Computer, and Systems Engineering*
*Rensselaer Polytechnic Institute*

**Reviewed on OpenReview:** *https://openreview.net/forum?id=Q9AkJpfJks&nesting=2&sort=date-desc*

## Abstract

Jumping connections enable Graph Convolutional Networks (GCNs) to overcome over-smoothing, while graph sparsification reduces computational demands by selecting a sub-matrix of the graph adjacency matrix during neighborhood aggregation. Learning GCNs with graph sparsification has shown empirical success across various applications, but a theoretical understanding of the generalization guarantees remains limited, with existing analyses ignoring either graph sparsification or jumping connections. This paper presents the first learning dynamics and generalization analysis of GCNs with jumping connections using graph sparsification. Our analysis demonstrates that the generalization accuracy of the learned model closely approximates the highest achievable accuracy within a broad class of target functions dependent on the proposed sparse effective adjacency matrix $A^*$. Thus, graph sparsification maintains generalization performance when $A^*$ preserves the essential edges that support meaningful message propagation. We reveal that jumping connections lead to different sparsification requirements across layers. In a two-hidden-layer GCN, the generalization is more affected by the sparsified matrix deviations from $A^*$ of the first layer than the second layer. To the best of our knowledge, this marks the first theoretical characterization of jumping connections' role in sparsification requirements. We validate our theoretical results on benchmark datasets in deep GCNs.

## 1 Introduction

Graph neural networks (GNNs) outperform traditional machine learning techniques when learning graph-structured data, that comprises a collection of features linked with nodes and a graph representing the correlation of the features. As one of the most popular variants of GNN, Graph Convolutional Networks (GCNs) (Kipf & Welling, 2017) perform the convolution operations on graphs by aggregating neighboring nodes to update the feature presentation of every node. GCNs have demonstrated great empirical success such as text classification (Norcliffe-Brown et al., 2018; Zhang et al., 2018) and recommendation systems (Wu et al., 2019; Ying et al., 2018). Because GCNs are easy and computationally efficient to implement, they are the preferred choice for large-scale graph training Duan et al. (2022); Zhang et al. (2022).

As the depth of vanilla GCNs increases, there is a tendency for the node representations to converge toward a common value, a phenomenon known as "over-smoothing" (Li et al., 2018). A widely adopted mitigation approach is to incorporate jumping connections, which allow features to bypass intermediate layers and directly contribute to future layers' output (Li et al., 2019; Xu et al., 2018b). Moreover, jumping connections are shown to accelerate the training process (Xu et al., 2021). Jumping connections have thus become an essential component of GCNs.

Processing large-scale graphs can be computationally demanding, particularly when dealing with the recursive neighborhood integration in GCNs. To alleviate this computational burden, graph sampling or sparsification methods select a subset of nodes or edges from the original graph when computing neighborhood aggregation. Various graph sampling approaches have been developed, including node sampling methods like GraphSAGE (Hamilton et al., 2017), layer-wise sampling like FastGCN (Chen et al., 2018), subgraph sampling methods like Graphsaint (Zeng et al., 2020). The graph sparsification methods (Li et al., 2020; Chen et al., 2021; You et al., 2020; Liu et al., 2023) usually co-optimize weights and sparsified masks to find optimal sparse graphs and remove the task-irrelevant edges. Ioannidis et al. (2020); Zeng et al. (2020); Srinivasa et al. (2020) introduce simple pruning methods that remove less significant edges without needing complex iterative training.

Although GCNs have demonstrated superior empirical success, their theoretical foundation remains relatively underdeveloped. Some works analyze the expressive power of GCNs in terms of the functions they can represent Morris et al. (2019); Cong et al. (2021); Oono & Suzuki (2019); Xu et al. (2019); Chen et al. (2019), while some works characterize the generalization gap which measures the gap between the training accuracy and test accuracy Esser et al. (2021); Liao et al. (2020); Tang & Liu (2023b). All these works ignore training dynamics, i.e., they do not characterize how to train a model to achieve great expressive power or a small generalization gap. Some works exploit the neural tangent kernel (NTK) technique to analyze the training dynamics of stochastic gradient descent and generalization performance simultaneously Yang et al. (2023). These analyses apply to deep neural networks, only when the network is impractically overparameterized, i.e., the number of neurons is either infinite Du et al. (2019a) or polynomial in the total number of nodes Qin et al. (2023).

Li et al. (2022a); Zhang et al. (2023b); Tang & Liu (2023a) analyze the training dynamics of SGD for GCNs with sparsification and prove that the learned model is guaranteed to achieve desirable generation. These analyses focus on two-hidden-layer GCNs, but the learning problem is already a nonconvex problem for these shallow networks. However, their network architectures exclude jumping connections. Xu et al. (2021) investigates training dynamics of jumping connections in multi-layer GCNs, but that paper does not provide generalization results and only considers linear activation functions.

To the best of our knowledge, this paper provides the first theoretical analysis of the training dynamics and generalization performance for two-hidden-layer GCNs with jumping-connection using graph sparsification. Our method focuses on the interaction of the jumping-connection and the intermediate layer and explains how jumping-connection influences training and graph sparsification across layers. We consider the semi-supervised node regression problem, where given all node features and partial labels, the objective is to predict the unknown node labels. Our major results include:

(1) We analyze training two-hidden-layer GCNs by stochastic gradient descent (SGD) with a jumping connection, using a pruning method that prefers the large weight edges from the adjacency matrix $A$. Our analysis demonstrates that the generalization accuracy of the learned model approximates the highest achievable accuracy within a broad class of target functions, which map input features to labels. Each target function is a sum of a simpler base function that contributes significantly to the output and a more complicated composite function that has a comparatively smaller impact on the output. This class encompasses a wide range of functions, including two-hidden-layer GCNs with jumping connections.

(2) This paper extends the concept of the sparse effective adjacency matrix, denoted as $A^*$, which is first introduced in Li et al. (2022a) for GCNs lacking jumping connections. This paper shows that $A^*$ also characterizes the influence of graph sparsification in GCNs with jumping connections. We find that the adjacency matrix $A$ of a graph often includes redundant information, suggesting that an effectively sparse graph can perform as well as or even surpass $A$ in training GCNs. Then the goal of graph pruning shifts

from minimizing the difference between sampled adjacency matrix, denoted by $A^s$, and $A$ to minimizing the difference between $A^s$ and $A^*$. Consequently, even when the pruned adjacency matrix $A^s$ is very sparse and significantly deviates from $A$, as long as there exists an $A^*$ closely enough to $A^s$, graph sparsification does not compromise the model's generalization performance.

(3) This paper theoretically demonstrates that, owing to the presence of jumping connections, sparsifying in different layers has different impacts on the model's output. Specifically, the first layer connects directly to the output through the jumping connection, and as a result, the deviation of the sampled matrix from the sparse effective matrix $A^*$ has a more significant effect than the deviation of the second layer. In contrast, the second layer influences the output through a composite function that contributes less significantly, allowing for more substantial deviations from $A^*$ in the process without compromising error rates. To the best of our knowledge, this is the first theoretical characterization of how jumping connections influence sparsification requirements, while previous analyses such as (Li et al., 2022a) assume that the sparsification approach remains consistent across different layers. Besides, our experiments on the deep-layer Jumping Knowledge Network GCN, demonstrate the significant impact of graph sampling in shallow layers compared to deeper layers. This empirical evidence supports our theoretical claims and the relevance of our two-layer model analysis in understanding deeper GCN architectures.

## 1.1 Related works

**Other theoretical analysis of GNNs** focus on expressive power and convergence analysis. Xu et al. (2018a); Morris et al. (2019) show the power of 1-hop message passing is upper bounded by 1-WL test. Feng et al. (2022); Wang & Zhang (2022) extend the analysis to k-hop message passing neural networks and spectral GNNs. Zhang et al. (2023a) explores the expressive power of GNNs from the perspective of graph biconnectivity. Oono & Suzuki (2020); Ramezani et al. (2020); Cong et al. (2021) investigates the optimization of GNN training.

**Generalization analyses of Neural Networks (NNs).** Various approaches have been developed to analyze the generalization of feedforward NNs. The neural tangent kennel (NTK) approach shows that overparameterized networks can be approximated by kernel methods in the limiting case (Jacot et al., 2018; Du et al., 2019b). The model estimation approach assumes the existence of a ground-truth one-hidden-layer model with desirable generalization and estimates the model parameters using the training data (Zhong et al., 2017; Zhang et al., 2020; Li et al., 2022b). The feature learning approach analyzes how a shallow NN learns important features during training and thus achieves desirable generalization (Li & Liang, 2018; Allen-Zhu & Li, 2022; 2023). All works ignore the jumping connection except Allen-Zhu & Li (2019), which analyzes the generalization of two-hidden-layer ResNet. Our analysis builds upon Allen-Zhu & Li (2019) and extends to GCNs with graph sparsification.

**Various GNN sparsification methods**. Node-wise sampling (Hamilton et al., 2017) randomly selects nodes and their multi-hop neighbors to create a localized subgraph. Layer-wise sampling (Chen et al., 2018; Zou et al., 2019; Huang et al., 2018) sample a fixed number of nodes in each layer. Subgraph-based sampling (Zeng et al., 2020; Chiang et al., 2019) generates subgraphs by sampling nodes and edges. As for graph sparsification, SGCN (Li et al., 2020) introduces the alternating direction method of multipliers (ADMM) to sparsify the adjacency matrix. UGS (Chen et al., 2021), Early-Bird (You et al., 2020), and ICPG (Sui et al., 2022) design a pruning strategy to sparsify the graph based on the lottery ticket hypothesis. CGP (Liu et al., 2023) proposes a graph gradual pruning framework to reduce the computational cost. DropEdge Rong et al. (2019), which randomly drops edges per layer during training.

## 2 Training GCNs with Layer-wise Graph Sparsification: Summary of Main Components

### 2.1 GCN Learning Setup

Let $\mathcal{G} = \{\mathcal{V}, \mathcal{E}\}$ represent an undirected graph, where $\mathcal{V}$ is the set of nodes with $|\mathcal{V}| = N$ nodes and $\mathcal{E}$ is the set of edges. $\Delta$ is the maximum degree of $\mathcal{G}$. An adjacency matrix $\tilde{A} \in R^{N \times N}$ is defined to describe the overall graph topology where $\tilde{A}(i, j) = 1$ if $(v_i, v_j) \in \mathcal{E}$ else $\tilde{A}(i, j) = 0$. $A$ denotes the normalized adjacency matrix with $A = D^{-\frac{1}{2}}(\tilde{A} + I)D^{-\frac{1}{2}}$ where $D$ is the degree matrix with diagonal elements $D_{i,i} = \sum_j \tilde{A}(i, j)$.

Each element $A_{ij}$ of the matrix $A$ represents the normalized weight of the edge between nodes $i$ and $j$. $a_n$ denotes the $n$th column of the matrix $A$. Let $X \in \mathbb{R}^{d \times N}$ denote the feature matrix of $N$ nodes, where $\tilde{x}_n \in \mathbb{R}^d$ denotes the feature of node $n$. Assume $\|\tilde{x}_n\| = 1$ for all $n$ without loss of generality. $y_n \in \mathbb{R}^k$ represents the label of node $n$. Let $\Omega \subset \mathcal{V}$ denote the set of labeled nodes. Given $X$ and partial labels in $\Omega$, the objective of semi-supervised node-regression is to predict the unknown labels in $\mathcal{V} \setminus \Omega$.

We consider training a two-hidden-layer GCN with a single jumping connection, where the function out : $\mathbb{R}^{d \times N} \times \mathbb{R}^{N \times N} \to \mathbb{R}^{k \times N}$ with

$$\text{out}(X, A; W, U) = C\sigma(WXA) + C\sigma(U\sigma(WXA)A) \tag{1}$$

where $\sigma(\cdot)$ applies the ReLU activation $\text{ReLU}(x) = \max(x, 0)$ to each entry, $W \in \mathbb{R}^{m \times d}$, $U \in \mathbb{R}^{m \times m}$, and $C \in \mathbb{R}^{k \times m}$ denote the first hidden-layer, second hidden-layer, and output layer weights, respectively. We only train $W$ and $U$. The output of the $n$th node can be written as $\text{out}_n : \mathbb{R}^{d \times N} \times \mathbb{R}^N \to \mathbb{R}^k$ is

$$\text{out}_n(X, A; W, U) = C\sigma(WXa_n) + C\sigma(U\sigma(WXA)a_n) \tag{2}$$

We focus on the $\ell_2$ regression task and the prediction loss of the $n$th node is defined as

$$\text{Obj}_n(X, A, y_n; W, U) = \frac{1}{2}\|y_n - \text{out}_n(X, A; W, U)\|_2^2 \tag{3}$$

The learning problem solves the following empirical risk minimization problem:

$$\min_{W,U} L_\Omega(W, U) = \frac{1}{|\Omega|} \sum_{n \in \Omega} \text{Obj}_n(X, A, y_n; W, U) \tag{4}$$

## 2.2 Training with stochastic gradient descent and graph sparsification

The recursive neighborhood aggregation through multiplying the feature matrix with $A$ is costly in both computation and memory. Graph sparsification prunes the graph adjacency matrix $A$ to reduce the computation and memory requirement. For example, one common theme of various edge sampling or sparsification methods is to retain the large weight edges $A_{ij}$ from the adjacency matrix $A$ in $A^s$ while pruning small weights (Chen et al., 2018; Zeng et al., 2020). To further reduce the computation, layer-wise sampling is also employed that uses different sampling rates in different layers, see, e.g., Chen et al. (2018).

We allow the sparsification methods with different parameter settings in different layers. Specifically, in the $t$th iteration, let $A^{1t}$ and $A^{2t}$ denote the sparsified adjacency matrix $A$ in the first and second hidden layers, respectively.

In Algorithm 1, (4) is solved by the stochastic gradient descent (SGD) method starting from random initialization. In each iteration, the gradient of the prediction loss of one randomly selected node is used to approximate the gradient of $L_\Omega$. Let $W^{(t)}$ and $U^{(t)}$ denote the current estimation of $W$ and $U$. When computing the stochastic gradient, instead of (2), we use[1]

$$\text{out}(X, A^{1t}, A^{2t}; W^{(t)}, U^{(t)}) = C\sigma(W^{(t)}XA^{1t}) + C\sigma(U^{(t)}\sigma(W^{(t)}XA^{1t})A^{2t}) \tag{5}$$

The main notations are summarized in Table 1 in Appendix.

# 3 Main Algorithm and Theoretical Results

## 3.1 Informal Key Theoretical Findings

We first summarize our major theoretical insights and takeaways before formally presenting them.

1. **The first theoretical generalization guarantee of two-hidden-layer GCNs with jumping-connection.** We demonstrate that training a single jumping-connection two-hidden-layer GCN using

---

[1]If different layers use different adjacency matrices, we specify both matrices in the function representation. Otherwise, we use one matrix to simplify notations.

our Algorithm 1 returns a model that achieves the label prediction performance almost the same as the best prediction performance using a large class of target functions. We also characterize quantitatively the required number of labeled nodes, referred to as the sample complexity, to achieve the desirable prediction error. To the best of our knowledge, only Li et al. (2022a); Zhang et al. (2023b) provide explicit sample complexity bounds for node classification using graph neural networks, but for shallow GCNs with no jumping connection. Our work is the first one that provides a theoretical generalization and sample complexity analysis for the practical GCN architecture with jumping connections.

2. **Graph sparsification affects generalization through the sparse effective adjacency matrix $A^*$.** We show that training with edge pruning produces a model with the same prediction accuracy as a model trained on a GCN with $A^*$ as the normalized adjacency matrix, i.e., replacing $A$ with $A^*$ in (1). The effective adjacency matrix is first discussed in (Li et al., 2022a), in the setup of node sampling for two-hidden-layer GCNs with no jumping connection, but $A^*$ in (Li et al., 2022a) is dense. We show that the effective adjacency matrix also exists for edge pruning on GCNs with jumping connection and can be sparse, indicating that the sparsified matrices can be very sparse without sacrificing generalization.

3. **Layer-wise graph sparsification due to jumping connection**. We show that in the two-hidden-layer GCN with a single jumping connection, the first hidden-layer learns a simpler base function that contributes more to the output, while the second hidden-layer learns a more complicated function that contributes less to the output. Therefore, compared with the first hidden layer, the second hidden layer is more robust to graph sparsification and can tolerate a deviation of the pruned matrix to $A^*$ without affecting the prediction accuracy. To the best of our knowledge, this is the first theoretical characterization of how jumping connections affect the sparsification requirements in different layers, while the previous analysis in (Li et al., 2022a) assumes the same matrix sampling deviations for all layers.

### 3.2 Graph Topology Sparsification Strategy

Our theoretical sparsification strategy differs slightly from Algorithm 1 due to our adjustments aiming to facilitate and simplify the theoretical analysis. Nevertheless, our core concept is remaining more large-weight edges with a higher probability, while remaining small-weight edges with a smaller probability.

We follow the same assumption on node degrees as that in Li et al. (2022a). Specifically, the node degrees in $\mathcal{G}$ can be divided into $L$ ($L \geq 1$) groups, with each group having $N_l$ nodes ($l \in [L]$). The degrees of all $N_l$ nodes in group $l$ are in the order of $d_l$, i.e., between $cd_l$ and $Cd_l$ for some constants $c \leq C$. $d_l$ is order-wise smaller than $d_{l+1}$, i.e., $d_l = o(d_{l+1})$.

Let $p_{ij}^k \in [0, \frac{1}{2}]$ ($k = 1, 2$) denote the probability of pruning the larger entries in $A_{B_{ij}}$, i.e., the larger entries are retained with probability $1 - p_{ij}^k$. A smaller $p_{ij}^k$ corresponds to more conservative pruning in retaining larger entries, while a larger $p_{ij}^k$ corresponds to a more aggressive pruning.

- *Sampling procedure:* Our pruning strategy operates block-wise. At each iteration, for each submatrix $A_{B_{ij}}$, we retain the top entries as follows:

Case (1) If $i > j$, each of the top[2] $d_1 \sqrt{\frac{d_i}{d_j}}$ largest entries $A_{ij}$ in $A_{B_{ij}}$ is retained independently with high probability $1 - p_{ij}^k$. The remaining entries in $A_{B_{ij}}$ are retained independently with a probability of $p_{ij}^k$.

Case (2) If $i \leq j$, each of the top $d_1$ largest entries $A_{ij}$ in $A_{B_{ij}}$ is retained with probability $1 - p_{ij}^k$. The remaining entries in $A_{B_{ij}}$ are retained independently with a probability of $p_{ij}^k$.

- *Layer-wise flexibility:* Although we consider a simplified sampling strategy for theoretical tractability and clarity, our framework allows pruning rates to vary across layers. This reflects the same core idea as practical layer-wise sampling methods Rong et al. (2019); Zeng et al. (2020); Chiang et al. (2019). Moreover, our sampling strategy differs from the theoretical analysis in Li et al. (2022a), where we adopt edge-level

---

[2]The values $d_1 \sqrt{\frac{d_i}{d_j}}$ and $d_1$ for selecting the top largest entries in $A_{ij}$ are chosen to simplify our theoretical analysis. In fact, any values in these orders are sufficient for our theoretical analysis. Note that the main idea of retaining lower-degree edges with higher probability is maintained in our sparsification strategy.

---

**Algorithm 1** SGD with Layer-wise Sparsification (LWS)

---

1: **Input:** Graph $\mathcal{G}$ with normalized adjancey matrix $A$, node features $X$, known labels in $\Omega$, step size $\eta_w$ and $\eta_v$, number of iterations $T$, pruning rate $p_{ij}^1$ and $p_{ij}^2$.

2: Initialize $W^{(0)}$, $V^{(0)}$, $C$. $\mathbf{W}_0 = 0$, $\mathbf{V}_0 = 0$

3: **for** $t = 0, 1, \cdots, T-1$ **do**

4:     Retain the top $q_1$ fraction of the largest entries with probability $(1 - p_{ij}^1)$ and retain the remaining $1 - q_1$ fraction of smaller entries with probability $(p_{ij}^1)$ to get $A^{1t}$.

5:     Retain the top $q_2$ fraction of the largest entries with probability $(1 - p_{ij}^2)$ and retain the remaining $1 - q_2$ fraction of smaller entries with probability $(p_{ij}^2)$ to get $A^{2t}$. ($q_1 > q_2$ and $p_{ij}^1 < p_{ij}^2$).

6:     Randomly sample $n$ from $\Omega$.

7:     Calculate the gradient of $L$ in (27) and update weight deviations through

$$\mathbf{W}_{t+1} \leftarrow \mathbf{W}_t - \eta_w \frac{\partial L(\mathbf{W}, \mathbf{V})}{\partial \mathbf{W}}\bigg|_{\mathbf{W}=\mathbf{W}_t, \mathbf{V}=\mathbf{V}_t}$$

$$\mathbf{V}_{t+1} \leftarrow \mathbf{V}_t - \eta_w \frac{\partial L(\mathbf{W}, \mathbf{V})}{\partial \mathbf{V}}\bigg|_{\mathbf{W}=\mathbf{W}_t, \mathbf{V}=\mathbf{V}_t}$$

8: **end for**

9: **Output:** $W^{(T)} = W^{(0)} + \mathbf{W}_T$, $V^{(T)} = V^{(0)} + \mathbf{V}_T$.

---

sampling with a layer-wise sparsification schedule, in contrast to their node-level sampling that is uniform across layers.

- *Degree-aware prioritization:* This sampling strategy inherently favors low-degree nodes. To see this, assume for simplicity that $p_{ij}^k < 1/2$ is the same across all groups. For two groups $j$ and $j'$ where group $j$ has a smaller degree, i.e., $d_j < d_{j'}$, we have $d_1\sqrt{\frac{d_i}{d_j}} > d_1\sqrt{\frac{d_i}{d_{j'}}}$. According to Case (1) of our sampling strategy, more entries in $A_{B_{ij}}$ than those in $A_{B_{ij'}}$ will be retained with probability $1 - p_{ij}^k$, while the remaining entries are kept with probability $p_{ij}^k < 1 - p_{ij}^k$. This means that more edges between groups $i$ and $j$ are preserved than edges between $i$ and $j'$. This introduces a structural bias toward retaining informative, low-degree connections.

To analyze the impact of this graph topology sparsification on the learning performance, we define the **sparse effective adjacency matrix** $A^*$ where in each submatrix $A_{B_{ij}}^*$:

(1) if $i > j$, the top $d_1\sqrt{\frac{d_i}{d_j}}$ largest values in $A_{B_{ij}}$ remain the same, while other entries are set to zero.

(2) if $i \leq j$, the top $d_1$ largest values in $A_{B_{ij}}$ remain the same, while other entries are set to zero.

One can easily check from the definition that $\|A^*\|_1 = O(1)$, i.e., the maximum absolute column sum of $A^*$ is bounded by a constant. Moreover, $A^*$ is sparse by definition.

### 3.3 Concept Class and Hierarchical Learning

In the context of GCNs, a concept class represents the set of possible target functions that map node features to labels. Defining this space is essential for understanding the function approximation capability of a learned GCN model and its ability to generalize to unseen data. Our theoretical generalization analysis establishes that the prediction error of the learned GCN model is bounded by a small constant multiple of the minimum achievable error within a well-defined concept class. This implies that the model effectively approximates the optimal function within this space. When the concept class accurately captures the true mapping from node features to labels, the minimum achievable prediction error approaches zero. Consequently, the learned GCN model also attains a low prediction error. This concept class depends on the sparsified adjacency matrix $A^*$ rather than the original $A$. We show that as long as the sparsified matrices $A^t$ remain close to $A^*$, even when highly sparse, graph sparsification does not degrade generalization performance.

To formally describe the concept class, consider a space of target functions $\mathcal{H}$, consisting of two smooth functions $\mathcal{F}$ and $\mathcal{G} : \mathbb{R}^{d \times N} \times \mathbb{R}^{N \times N} \to \mathbb{R}^{k \times N}$, along with a constant $\alpha \in \mathbb{R}^+$:

$$\mathcal{H}_{A^*}(X) = \mathcal{F}_{A^*}(X) + \alpha \mathcal{G}_{A^*}(\mathcal{F}_{A^*}(X)), \tag{6}$$

where the $r$-th row ($r \in [k]$) of $\mathcal{F}_{A^*}$ and $\mathcal{G}_{A^*}$, denoted by $\mathcal{F}^r$ and $\mathcal{G}^r : \mathbb{R}^{d \times N} \times \mathbb{R}^{N \times N} \to \mathbb{R}^{1 \times N}$, satisfy:

$$
\begin{aligned}
\mathcal{F}_{A^*}^r(X) &= \sum_{i=1}^{p_{\mathcal{F}}} a_{\mathcal{F},r,i}^* \cdot \mathcal{F}^{r,i}(w_{r,i}^{*T} X A^*), \\
\mathcal{G}_{A^*}^r(X) &= \sum_{i=1}^{p_{\mathcal{G}}} a_{\mathcal{G},r,i}^* \cdot \mathcal{G}^{r,i}(v_{r,i}^{*T} X A^*),
\end{aligned}
\tag{7}
$$

where $p_{\mathcal{F}}, p_{\mathcal{G}}$ are the counts of basis functions used to construct the decompositions of $\mathcal{F}^r$ and $\mathcal{G}^r$; $a_{\mathcal{F},r,i}^*, a_{\mathcal{G},r,i}^* \in [-1,1]$ are scalar coefficients for given $r, i$; $w_{r,i}^* \in \mathbb{R}^d$ and $v_{r,i}^* \in \mathbb{R}^k$ are vectors with norms $\|w_{r,i}^*\| = \|v_{r,i}^*\| = \frac{1}{\sqrt{2}}$ for all $r, i$; $\mathcal{F}^{r,i}, \mathcal{G}^{r,i} : \mathbb{R} \to \mathbb{R}$ are smooth activation functions applied element-wise.

The complexities of $\mathcal{F}$ and $\mathcal{G}$ are represented by the tuples $(p_{\mathcal{F}}, \mathcal{C}_s(\mathcal{F}), \mathcal{C}_m(\mathcal{F}, error))$ and $(p_{\mathcal{G}}, \mathcal{C}_s(\mathcal{G}), \mathcal{C}_m(\mathcal{G}, error))$, respectively. $\mathcal{C}_m$ and $\mathcal{C}_s$ represent model and sample complexities, respectively. The overall complexity of $\mathcal{H}$ is quantified by the tuple $(p_{\mathcal{F}}, p_{\mathcal{G}}, \mathcal{C}_s(\mathcal{F}), \mathcal{C}_s(\mathcal{G}), \mathcal{C}_m(\mathcal{F}, error), \mathcal{C}_m(\mathcal{G}, error))$. The complexity of $\mathcal{F}$ (or $\mathcal{G}$) is determined by the most complex sub-target function among the $p_{\mathcal{F}}$ (or $p_{\mathcal{G}}$) smooth functions. Specifically, the complexities for $\mathcal{F}$ and $\mathcal{G}$ are defined as:

$$
\begin{aligned}
\mathcal{C}_m(\mathcal{F}, error) &= \max_{r,i} \left\{ \mathcal{C}_m(\mathcal{F}^{r,i}, error, \|A^*\|_1) \right\}, \quad \mathcal{C}_s(\mathcal{F}) = \max_{r,i} \left\{ \mathcal{C}_s(\mathcal{F}^{r,i}, \|A^*\|_1) \right\}, \\
\mathcal{C}_m(\mathcal{G}, error) &= \max_{r,i} \left\{ \mathcal{C}_m(\mathcal{G}^{r,i}, error, \|A^*\|_1) \right\}, \quad \mathcal{C}_s(\mathcal{G}) = \max_{r,i} \left\{ \mathcal{C}_s(\mathcal{G}^{r,i}, \|A^*\|_1) \right\}.
\end{aligned}
\tag{8}
$$

The model and sample complexity definitions follow similarly to those in Li et al. (2022a) (Section 1.2) and Allen-Zhu & Li (2019) (Section 4). Please see Appendix B for details.

$\mathcal{F}$ and $\mathcal{G}$ can both be viewed as one-hidden-layer GCNs with smooth activation functions and adjacency matrix $A^*$. The target function $\mathcal{H}$ includes the base signal $\mathcal{F}$, which is less complex yet contributes significantly to the target, and $\mathcal{G}$, which is more complicated but contributes less. We will show that the learner networks defined in (5) can learn the concept class of target functions defined in (6). Intuitively, we will show that using a two-hidden-layer GCN with a jumping connection, the first hidden layer learns the low-complexity $\mathcal{F}$, and the second hidden layer learns the high-complexity $\mathcal{G}(\mathcal{F})$ with the help of $\mathcal{F}$ learned by the first hidden layer using the jumping connection.

We will also show that the learned GCN by our method performs almost the same as the best function in the concept class in predicting unknown labels. Let $\mathcal{D}_{\tilde{x}_n}$ and $\mathcal{D}_{y_n}$ denote the distribution from which the feature and label of node $n$ are drawn, respectively. Let $\mathcal{D}$ denote the concatenation of these distributions. Then the given feature matrix $X$ and partial labels in $\Omega$ can be viewed as $|\Omega|$ identically distributed but correlated samples $(X, y_n)$ from $\mathcal{D}$. The correlation results from the fact that the label of node $i$ depends on not only the feature of node $i$ but also neighboring features.

The $n$-th column of $\mathcal{H}_{A^*}$, denoted $\mathcal{H}_{n,A^*} : \mathbb{R}^{d \times N} \times \mathbb{R}^{N \times N} \to \mathbb{R}^k$, represents the target function for node $n$. To measure the label approximation performance of the target function, define

$$\mathbb{E}_{(X,y_n) \sim \mathcal{D}, n \in \mathcal{V}} \left[ \frac{1}{2} \|\mathcal{H}_{n,A^*}(X) - y_n\|_2^2 \right] = \text{OPT} \tag{9}$$

as the minimum prediction error achieved by the best target function in the concept class in (8). OPT decreases when the target functions are more complex, or the concept class enlarges, or if $A^*$ characterizes the node correlations properly.

### 3.4 Modeling the prediction error of unknown labels

To simplify the analysis and representation, we re-parameterize $U$ in (1) and (2) as $VC$, where $V \in \mathbb{R}^{m \times k}$. Then, (2) can be rewritten as follows:

$$
\begin{aligned}
\text{out}_n(X, A; W, V) &= \text{out}_n^1(X, A) + C\sigma(V \text{ out}^1(X, A) a_n) \\
\text{where out}^1(X, A; W) &= C\sigma(WXA), \\
\text{out}_n^1(X, A; W) &= C\sigma(WXa_n)
\end{aligned}
\tag{10}
$$

We follow the conventional setup for theoretical analysis that $C$ is fixed at its random initialization, and only $W$ and $V$ are updated during training. $C$, $W^{(0)}$, $V^{(0)}$ are randomly initialized from Gaussian distributions, $C_{i,j} \overset{\text{i.i.d.}}{\sim} \mathcal{N}\left(0, \frac{1}{m}\right)$, $W_{i,j}^{(0)} \overset{\text{i.i.d.}}{\sim} \mathcal{N}\left(0, \sigma_w^2\right)$ and $V_{i,j}^{(0)} \overset{\text{i.i.d.}}{\sim} \mathcal{N}\left(0, \sigma_v^2/m\right)$, respectively.

The algorithm is summarized in Algorithm 1. When computing the stochastic gradient of a sampled label $y_n$, the loss is represented as a function of the weight deviations $\mathbf{W}, \mathbf{V}$ from initiation $W^{(0)}$ and $V^{(0)}$, i.e.,

$$L(\mathbf{W}, \mathbf{V}) = \text{Obj}_n(X, A^{1t}, A^{2t}, y_n; W^{(0)} + \mathbf{W}, V^{(0)} + \mathbf{V}). \tag{11}$$

$\mathbf{W}_t$ and $\mathbf{V}_t$ denote the weight deviations of the estimated weights $W^{(t)}$ and $V^{(t)}$ in iteration $t$ from $W^{(0)}$ and $V^{(0)}$, i.e., $W^{(t)} = W^{(0)} + \mathbf{W}_t$, $V^{(t)} = V^{(0)} + \mathbf{V}_t$. We assume $0 < \alpha \le \widetilde{O}\left(\frac{1}{\mathcal{C}_s(\mathcal{G})}\right)$ throughout the training. We prove that $\|\mathbf{W}_t\|_2$ and $\|\mathbf{V}_t\|_2$ are bounded by $\widetilde{\Theta}\left(\mathcal{C}_s(\mathcal{F})\right)$ and $\widetilde{\Theta}\left(\alpha \mathcal{C}_s(\mathcal{G})\right)$ during training, i.e., $\|\mathbf{W}_t\|_2 \le \widetilde{\Theta}\left(\mathcal{C}_s(\mathcal{F})\right)$, $\|\mathbf{V}_t\|_2 \le \widetilde{\Theta}\left(\alpha \mathcal{C}_s(\mathcal{G})\right) < 1$ for all $t$.

The following lemma shows that graph sparsification in different layers contributes to the output approximation differently. In other words, to maintain the same accuracy in the output, the tolerable pruning rates in different layers are different.

**Lemma 3.1.** *For any given constant $E$, if the first and second layer matrices $A^{1t}$ and $A^{2t}$ are sparsified with probabilities satisfying*

$$p_{ij}^1 \le \widetilde{\Theta}\left(\frac{\sqrt{d_i d_j}E}{N_i N_j \mathcal{C}_s(\mathcal{F})}\right), \quad p_{ij}^2 \le \widetilde{\Theta}\left(\frac{\sqrt{d_i d_j}E}{N_i N_j \alpha \mathcal{C}_s(\mathcal{F})\mathcal{C}_s(\mathcal{G})}\right), \tag{12}$$

*then with probability at least $1 - e^{-\Omega(E\sqrt{d_i d_j}/\mathcal{C}_s(\mathcal{F}))}$, we have*

$$\left\|A^{1t} - A^*\right\|_1 \le \frac{E}{\widetilde{\Theta}(\mathcal{C}_s(\mathcal{F}))}, \tag{13}$$

$$\left\|A^{2t} - A^*\right\|_1 \le \frac{E}{\widetilde{\Theta}(\alpha \mathcal{C}_s(\mathcal{F})\mathcal{C}_s(\mathcal{G}))}, \tag{14}$$

$$\left\|\text{out}_n(X, A^{1t}, A^{2t}; W^{(t)}, V^{(t)}) - \text{out}_n(X, A^*; W^{(t)}, V^{(t)})\right\|_2 \le E. \tag{15}$$

Lemma 3.1 demonstrates that skip connections enable more flexible sparsification in deeper layers. To see this, note that since $\widetilde{\Theta}(\alpha \mathcal{C}_s(\mathcal{G})) < 1$ (see Table 3 in the Appendix), the second-layer sparsification condition in (12) permits larger values of $p_{ij}^2$ compared to $p_{ij}^1$, allowing more aggressive pruning in the second layer while still satisfying the output error bound in (15). This is because the skip connection ensures that the final output aggregates features from each layer in a decoupled manner, allowing the approximation error at each layer (see (13) and (14)) to be independently controlled.

We will show the learned model can achieve an error close to $O(\text{OPT})$. Our main theorem can be sketched as follows,

**Theorem 3.2.** *For $\alpha \in \left(0, \widetilde{\Theta}\left(\frac{1}{\mathcal{C}_s(\mathcal{G})}\right)\right)$, let $\epsilon_0 = \widetilde{\Theta}(\alpha^4 \mathcal{C}_s(\mathcal{G})^4) < 1$ and define the target error $\epsilon = 10 \cdot \text{OPT} + \epsilon_0$. Suppose the pruning probabilities $p_{ij}^1, p_{ij}^2$ satisfy (12) with $E = \epsilon_0$. Then there exist $M_0 = \text{poly}\left(\mathcal{C}_m(\mathcal{F}, \alpha), \mathcal{C}_m(\mathcal{G}, \alpha), \|A^*\|_1, \alpha^{-1}\right)$, $T_0 = \widetilde{\Theta}\left(\frac{\mathcal{C}_s(\mathcal{F})^2}{\|A^*\|_1 \min\{0.1, \epsilon^2\}}\right)$, $N_0 = \widetilde{\Theta}\left(\Delta^4 \mathcal{C}_s(\mathcal{F})^2 \|A^*\|_1^4 \log N \cdot \epsilon^{-2}\right)$, such that for any $m \ge M_0$, $T \ge T_0$, and $|\Omega| \ge N_0$, with high probability over the random initialization and training process, the SGD algorithm satisfies:*

$$\frac{1}{T}\sum_{t=0}^{T-1} \mathbb{E}_{(X, y_n)\sim \mathcal{D}, n\in\mathcal{V}} \left\|y_n - \text{out}_n\left(X, A^{1t}, A^{2t}; \mathbf{W}_t, \mathbf{V}_t\right)\right\|_2^2 \le \epsilon. \tag{16}$$

(16) shows that the learned GCN achieves a prediction error no worse than $\epsilon$, averaged over the data distribution and training iterations. Note that when the concept class becomes more expressive, $\mathcal{C}_m$ and $\mathcal{C}_s$

increase, while the optimal error OPT decreases. According to Theorem 3.2, this leads to an increase in the required the model complexity $M_0$ (the number of neurons) and the sample complexity $N_0$ (the number of labeled nodes), while the generalization error epsilon decreases. Thus, as a sanity check, our theoretical bounds match the intuition that a larger model and more labels improve the prediction accuracy.

In parallel, the pruning probabilities $p_{ij}^1$ and $p_{ij}^2$ are positively correlated with $\epsilon_0$, indicating that achieving a smaller generalization error requires lower pruning probabilities, i.e., more conservative pruning improves generalization. Between the two, $p_{ij}^2$ is consistently larger than $p_{ij}^1$, suggesting that more aggressive pruning is permissible in the second layer while maintaining conservative pruning in the first. Moreover, $N_0 = \widetilde{\Theta}(\log N)$ suggests that a logarithmic number of labels suffices to generalize to the entire graph under our assumptions. Finally, as $\|A^*\|_1$ increases, the constants $C_m$ and $\mathcal{C}_s$ also increase, which in turn raises $M_0$ and $N_0$. This reflects a natural phenomenon: when the affective adjacency matrix becomes denser, the prediction task becomes harder, and generalization performance degrades accordingly.

This proof of Theorem 3.2 builds upon the proof of Theorem 1 in Allen-Zhu & Li (2019), which analyzes the generalization of a three-layer ResNet for a supervised regression problem. We extend the analysis to training GCNs with graph sparsification for a semi-supervised node regression problem. The main technical challenge is to handle the dependence of labels on neighboring features and the error in adjacency matrices due to the sparsification. Compared with Li et al. (2022a) which also considers training GCN with graph sampling, we consider a different sparsification method from that in Li et al. (2022a). The resulting $A^*$ in Li et al. (2022a) is a dense matrix as $A$, while $A^*$ in our paper is a sparse matrix. Our results thus allow the sparsified matrices to be very sparse while still maintaining the generalization accuracy. Moreover, the sampling method is the same for both hidden layers in Li et al. (2022a), resulting the same deviation from $A^*$ in both layers. Our results indicate that the jumping connection allows a more flexible sparsification method in the second layer.

### 3.5 Proof Overview

In practice, for computational efficiency, we use the sparsified adjacency matrix $A^t$ in the learning network. Therefore, the discrepancy between the target function with $A^*$ and the practical learning network with $A^t$ can be viewed as two parts:

$$
\begin{aligned}
\left\|\mathcal{H}_{n,A^*}(X) - \text{out}_n\left(X, A^{1t}, A^{2t}; \mathbf{W}_t, \mathbf{V}_t\right)\right\|_2 \leq &\left\|\mathcal{H}_{n,A^*}(X) - \text{out}_n\left(X, A^*\right)\right\|_2 \\
&+ \left\|\text{out}_n\left(X, A^{1t}, A^{2t}\right) - \text{out}_n\left(X, A^*\right)\right\|_2
\end{aligned}
\tag{17}
$$

the first part quantifies how well the learning network, trained with $\mathbf{W}_t$ and $\mathbf{V}_t$ using $A^*$, can approximate the target function $\mathcal{H}_{n,A^*}$. We prove the existence of $\mathbf{W}^*$ and $\mathbf{V}^*$ (see Lemma C.3) Problem statementwith $m \geq M_0, TstatementT_0$ and $\Omega \geq N_0$ such that

$$
\left\|\mathcal{H}_{n,A^*}(X) - \text{out}_n\left(X, A^*; \mathbf{W}^*, \mathbf{V}^*\right)\right\|_2 \leq \epsilon_0. -
\tag{18}
$$

The second part quantifies the difference between the learning network's output when using the sparse adjacency matrices $A^t$ and effective adjacency matrix $A^*$. Specifically, it is represented by the term:

$$
\begin{aligned}
\left\|\text{out}_n\left(X, A^{1t}, A^{2t}\right) - \text{out}_n\left(X, A^*\right)\right\|_2 \leq &\left\|C\sigma(WXa_n^{1t}) - C\sigma(WXa_n^*)\right\|_2 + \\
&\left\|C\sigma(V\,\text{out}_n^1(XA^{1t})a_n^{2t}) - C\sigma(V\,\text{out}_n^1(XA^*)a_n^*)\right\|_2
\end{aligned}
\tag{19}
$$

For the inequality $\left\|C\sigma(WXa_n^{1t}) - C\sigma(WXa_n^*)\right\|_2 \leq \epsilon_0$ to hold, it is required that $\left\|A^{1t} - A^*\right\|_1 \leq \frac{\epsilon_0}{\widetilde{\Theta}(\mathcal{C}_s(\mathcal{F}))}$ and similarly, $\left\|A^{2t} - A^*\right\|_1 \leq \frac{\epsilon_0}{\widetilde{\Theta}(\alpha\mathcal{C}_s(\mathcal{F})\mathcal{C}_s(\mathcal{G}))}$ (see Appendix C.4). We establish that with appropriate pruning probabilities $p_{ij}^1$ and $p_{ij}^2$, the norms $\left\|A^{1t} - A^*\right\|_1$ and $\left\|A^{2t} - A^*\right\|_1$ can be sufficiently small (see Appendix C.7).

Finally, consider the definition of OPT, we can prove $\left\|y_n - \text{out}_n\left(X, A^{1t}, A^{2t}; \mathbf{W}_t, \mathbf{V}_t\right)\right\|_2 \leq \epsilon$.

## 4  Empirical Experiment

### 4.1  Experiment on synthetic data

We generate a graph $\mathcal{G}$ with $N = 2000$ nodes. Given $A$, the sparse $A^*$ is obtained following the procedure in Section 3.2. The node labels are generated by the target function

$$
\begin{aligned}
\mathcal{F}_{A^*}(X) &= C W^* X A^*, \\
\mathcal{G}_{A^*}(\mathcal{F}_{A^*}(X)) &= C\left(\sin\left(V^* \mathcal{F}_{A^*}(X) A^*\right) \odot \tanh\left(V^* \mathcal{F}_{A^*}(X) A^*\right)\right), \\
\mathcal{H}_{A^*}(X) &= \mathcal{F}_{A^*}(X) + \alpha \mathcal{G}_{A^*}(\mathcal{F}_{A^*}(X)).
\end{aligned}
\tag{20}
$$

where $X \in \mathbb{R}^{d \times N}$, $W^* \in \mathbb{R}^{r \times d}$, $V^* \in \mathbb{R}^{r \times k}$, and $C \in \mathbb{R}^{k \times r}$ are randomly generated with each entry i.i.d. from $\mathcal{N}(0, 1)$. $d = 100$, $k = 5$, $r = 30$, $\alpha = 0.5$. A two-hidden-layer GCN with a single jumping connection as defined in (2) with $m$ neurons in each hidden layer is trained on a randomly selected set $\Omega$ of labeled nodes. The rest $N - |\Omega|$ labels are used for testing. The test error is measured by the $\ell_2$ regression loss in (3).

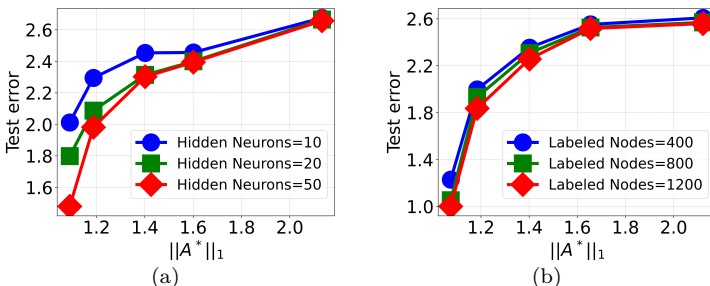

Figure 1: Experiment on two-degree group synthetic data: To achieve the same test error, if $\|A^*\|_1$ is larger, (a) then the number of neurons $m$ is larger (when $|\Omega| = 1200$ is fixed); (b) then the number of labeled nodes is larger (when $m = 50$ is fixed).

**How $\|A^*\|_1$ impacts model and sample complexities**: In Figures 1, $\mathcal{G}$ has two-degree groups. Group 1 has $N_1 = 200$ nodes, and the degree of each node follows a Gaussian distribution $\mathcal{N}(d_1, \sigma^2)$. Group 2 has $N_2 = 1800$ nodes, and the degree of each node follows a Gaussian distribution $\mathcal{N}(d_2, \sigma^2)$. The degrees are truncated to fall within the range of 0 to 500. We vary $A^*$ by changing $d_2$ and the corresponding $A$. We fix $d_1 = 200$ and $\sigma = 20$. We directly train with $A^*$ to study the impact of $A^*$ on model and sample complexities. In Figures 1 (a), $|\Omega| = 1200$ and the number of neurons per layer $m$ varies. To achieve the same test accuracy, when $\|A^*\|_1$ increases, the number of neurons also increases, verifying our model complexity $M_0$ in Theorem 3.2. In Figures 1 (b), $m = 50$ and $|\Omega|$ varies. To achieve the same test accuracy, when $\|A^*\|_1$ increases, the required number of labels also increases, verifying our sample complexity $N_0$ in Theorem 3.2.

In Figures 2, $\mathcal{G}$ has one-degree groups and the degree of each node follows a Gaussian distribution $\mathcal{N}(d, \sigma^2)$. $d = 200$. The degrees are truncated to fall within the range of 0 to 500. We vary $A^*$ by changing $\sigma$ and the corresponding $A$. We directly train with $A^*$ to study the impact of $A^*$ on model and sample complexities. In Figures 2 (a), $|\Omega| = 1200$ and the number of neurons per layer $m$ varies. Fig. 2 (a) shows the testing error decreases as $m$ increases. When $m$ is the same, the testing error increases as $\|A^*\|_1$ increases. This verifies our model complexity in Theorem 3.2. In Figures 2 (b), $m = 50$ and $|\Omega|$ varies. Fig. 2 (b) shows the testing error decreases as $\Omega$ increases. When $\Omega$ is the same, the testing error increases as $\|A^*\|_1$ increases. This verifies our model complexity in Theorem 3.2.

**Layer-wise Sparsification impact on generalization**: We fix $\Omega = 1200$, $m = 50$, $\|A^*\|_1 = 1.27$. We sample adjacency matrices during training. Figure 3 shows the relationship between the test error and the average deviation of sparsified matrices ($A^{1t}$ and $A^{2t}$ in the first and second hidden layers) from $A^*$. We can see that pruning in the second hidden layer (blue dashed line) contributes to generalization degradation much milder than pruning in the first hidden layer (green solid arrow). This verifies our Lemma 3.1 that the sparsification requirements are more restrictive in the first layer than the second layer to maintain the same generalization accuracy.

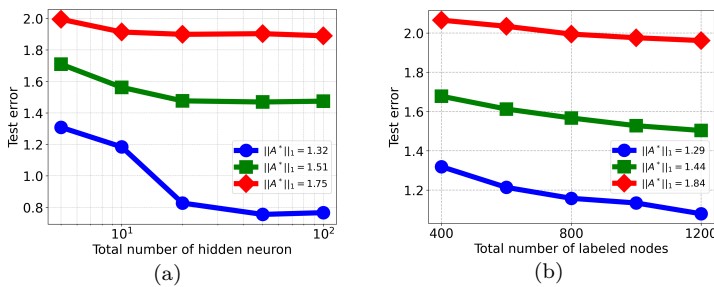

Figure 2: Experiment on one-degree group synthetic data: (a) Test error with $|\Omega| = 1200$. (b) Test error with $m = 50$. The test error increases as $\|A^*\|_1$ increases while others remain the same.

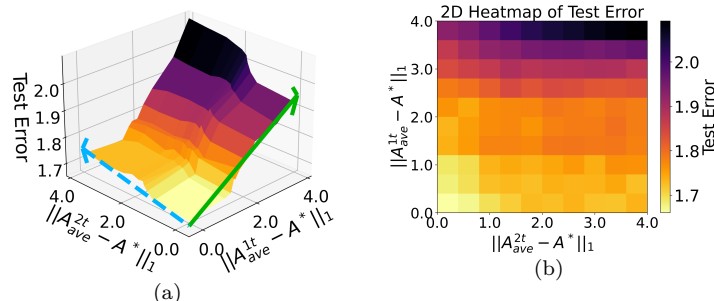

Figure 3: Experiment on synthetic data of layer-wise sparsification. (a) Sampling in the first hidden layer affects the test performance more significantly than the second hidden layer. (b) 2D heatmap of test error.

## 4.2 Experiments on Small-scale Real Datasets

**Retaining large weights in the graph can perform as well as trained sparse graph methods**. We applied Algorithm 1 to a one-hidden-layer shallow GCN on small-scale datasets (Cora, Citeseer) for node multi-class classification tasks, comparing it with state-of-the-art (SOTA) sparsification methods such as CGP (Liu et al., 2023) and UGS (Chen et al., 2021). For these small datasets, multi-layer GCNs are not necessary, so we employ the one-hidden-layer GCN (Kipf & Welling, 2017) with 512 hidden neurons and preclude the use of different pruning rates for shallow and deep layers. We retain the top $q$ fraction of the largest edge weights with a 99% probability and retain the remaining $1 - q$ fraction of small weight with a 1% probability to get $A^t$, so the sparsify of our method (LWS) is $0.98q + 0.01$ and we vary $q$ from 0.01 to 1.0. In Figure 4, we only demonstrate that retaining large-weight edges from $A$ using our method can achieve performance comparable to that of trained sparse graphs produced by SOTA methods.

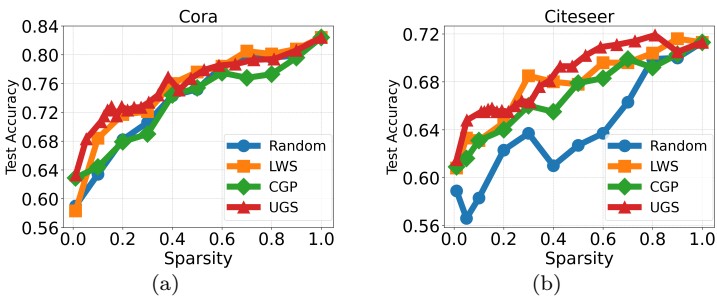

Figure 4: Experiment on sparsifying shallow GCN models.

### 4.3 Experiments on Large-scale Real Datasets

We also evaluate multi-layer GCNs with jumping connections on the large-scale Open Graph Benchmark (OGB) datasets for node multi-class classification tasks. A summary of Ogbn datasets' statistics is presented in Table 2 in Appendix. While our theory does not explicitly model dropout or normalization, we follow standard practice and include them in real-data experiments, where the theoretical insights remain valid.

The task of Ogbn-Arxiv is to classify the 40 subject areas of arXiv CS papers. We use 60% of the data for training, 20% for testing, and 20% for verification. We deploy an 8-layer Jumping Knowledge Network (Xu et al., 2018b) GCN with concatenation layer aggregation as a learner network. We treat the first four layers as shallow layers and the last four layers as deep layers. Shallow and deep layers are sparsified differently. The generalization is evaluated by the fraction of erroneous predictions of unknown labels.

**Pruning in deep layers is more flexible with less generalization degradation**. In this experiment, we employ a simplified version of the graph sparsification method discussed in Section 3.2. For the shallow layers, at each iteration $t$, we obtain a sparsified adjacency matrix $A^{1t}$ as follows: we retain the top $q_1$ fraction of the largest weight edges $A_{ij}$ from the adjacency matrix $A$ with a 99% probability, and retain the remaining entries with a 1% probability. For the deep layers, the sparsified adjacency matrix $A^{2t}$ is generated similarly, but we use the top $q_2$ fraction of largest $A_{ij}$, again retaining with probabilities of 99% and 1% for the top and remaining entries, respectively.

Figure 5 shows test error when $q_1$ and $q_2$ vary. One can see that the test error decreases more drastically when only increasing $q_1$ (blue dashed arrow) compared with only increasing $q_2$ (green solid arrow), indicating that graph pruning in shallow layers has a more significant impact than graph pruning in deeper layers. When both $q_1$ and $q_2$ are greater than 0.6, the test error is always small (less than 0.29) for a wide range of $q_1, q_2$. That may suggest the existence of multiple sparse $A^*$ such that sparsified matrices $A^t$ with different $q_1, q_2$ pairs approximate different $A^*$, and all $A^*$ can accurately represent the data correlations in the mapping function from the features to the labels.

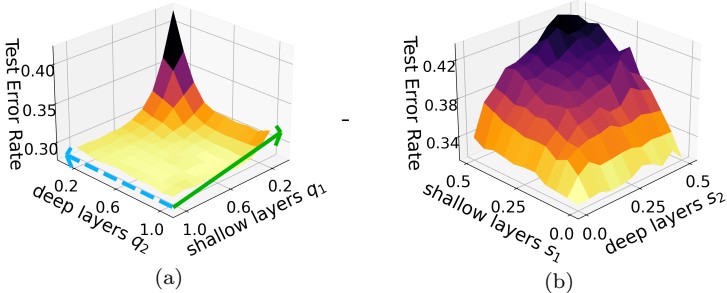

Figure 5: Learning deep GCNs on Ogbn-Arxiv: (a) Deeper layers tolerate higher sampling rates than shallow layers while maintaining accuracy. (b) 2D heatmap of test error rate.

**Large-weight edges are more influential on generalization than small-weight edges**. Note that if nodes $i$ and $j$ have higher degrees, then $A_{ij}$ has a smaller value. We sparsify one matrix for the shallow layers by keeping the values of $A_{ij}$ that are in the range of top $s_1$ to $s_1 + 0.5$ fraction and setting all other values to zero. Similarly, we sparsify one matrix for the deep layers by keeping the values in the range of top $s_2$ to $s_2 + 0.5$ fraction and setting all other values to zero. These two sparsified matrices are used during training. When $s_1$ and $s_2$ increase, the resulting matrices have the same number of nonzero entries, and the sparsified entries focus more on high-degree edges. Figure 6 shows the test error indeed increases as $s_1, s_2$ increases. This justifies the sparsification strategy to retain more large-weight edges.

For the Ogbn-Products dataset, we deploy a 4-layer Jumping Knowledge Network (Xu et al., 2018b) GCN with concatenation layer aggregation. Each hidden layer consists of 128 neurons. We define the first two layers as shallow layers and the last two layers as deep layers with sampling rate $p_2$. The task is to classify

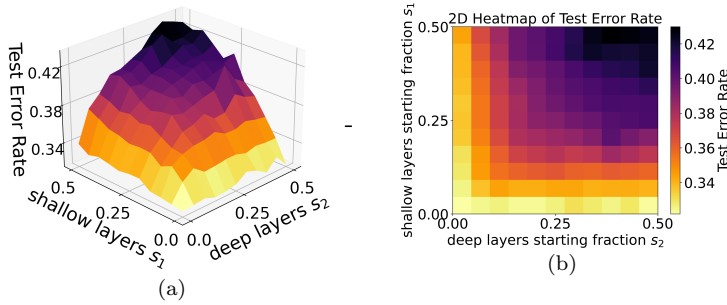

Figure 6: Learning deep GCNs on Ogbn-Arxiv: (a) Retaining more large-weight edges (small $s_1, s_2$) outperforms retaining more small-weight edges (large $s_1, s_2$). (b) 2D heatmap of test error rate.

the category of a product in a multi-class, where the 47 top-level categories are used for target labels. We use 60% of the data for training, 20% for testing, and 20% for verification.

We run the similar experiment as Figure 5 for Ogbn-Products dataset. We fix $q_1 = 0.1$ and vary $q_2$ from 0.1 to 1.0 at increments of 0.1, then fix $q_2 = 0.1$ and vary $q_1$ from 0.1 to 1.0 at increments of 0.1. Figure 7 shows that with the increasing sampling rate in shallow layers, the test accuracy is higher than the test accuracy with the increasing sampling rate in deep layers. It suggests that the generalization is more sensitive to the sampling in the shallow layers rather than deep layers, consistent with observations in other datasets.

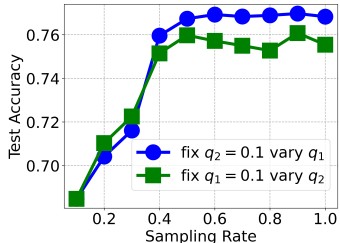

Figure 7: Layer-wise Sampling Rate Effect on Ogbn-Products

## 5 Conclusion and Future Work

This paper provides a theoretical generalization analysis of training GCNs with skip connections using graph sampling. To the best of our knowledge, this paper provides the first analysis of how skip connection affects the generalization performance. We show that for a two-hidden-layer GCN with a skip connection, the first hidden layer learns a simpler function that contributes significantly to the output, making the choice of sampling more crucial in the first hidden layer. In contrast, the second layer learns a composite function that contributes less to the output, allowing for a more flexible sampling approach while preserving generalization. This insight is verified on deep GCNs on benchmark datasets. Our analysis provides a general guideline: apply conservative sparsification in shallow layers to preserve local neighborhood information, and more aggressive pruning in deeper layers, especially when skip connections are present, to balance expressiveness and efficiency.

Future work includes extending our analysis to other graph neural networks and practical architectures. For example, our framework could be adapted to spatio-temporal GCNs by incorporating temporal edges into the sparsified adjacency matrix. Extending the analysis to attention-based models such as Graph Attention Networks (GATs) and Graph Transformers is another exciting direction, though it would require new tools to characterize dynamic attention mechanisms and global message passing. Our insight of layer-wise decoupling via skip connections naturally extend to deeper GNNs, though formalizing these insights in very deep architectures may require additional tools to handle cumulative sparsification effects and complex nonlinear interactions. Additionally, understanding the interplay of skip connections with practical components

such as dropout and normalization layers remains an open problem, as these mechanisms introduce distinct theoretical challenges due to their stochasticity and feature-rescaling effects.

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

## A    Preliminaries

We first restate some important notations used in the Appendix, which are summarized in Table 1.

Table 1: Summary of Notations

| Notations | Annotation |
|---|---|
| $\mathcal{G} = \{\mathcal{V}, \mathcal{E}\}$ | $\mathcal{G}$ is an undirected graph consisting of a set of nodes $\mathcal{V}$ and a set of edges $\mathcal{E}$. |
| $N$ | The total number of nodes in a graph. |
| $A = D^{-\frac{1}{2}} \tilde{A} D^{-\frac{1}{2}}$ | $A \in \mathbb{R}^{N \times N}$ is the normalized adjacency matrix computed by the degree matrix $D$ and the initial adjacency matrix $\tilde{A}$. |
| $A^*$ | The effective adjacency matrix. |
| $A^{1t}, A^{2t}$ | The sparsified adjacency matrices $A$ in the first and second hidden layers at the $t$-th iteration, respectively. |
| $M_0$ | The required number of neurons (model complexity). |
| $T_0$ | The required number of iterations for convergence in the SGD algorithm. |
| $N_0$ | The required number of labeled samples (sample complexity). |
| $W \in \mathbb{R}^{m \times d}$ | The weight matrix for the first hidden layer. |
| $U \in \mathbb{R}^{m \times m}$ | The weight matrix for the second hidden layer. |
| $C \in \mathbb{R}^{k \times m}$ | The weight matrix for the output layer. |
| $V \in \mathbb{R}^{m \times k}$ | The re-parameterized weight matrix used in place of $U$ in the second hidden layer. |
| $a_n \in \mathbb{R}^N$ | The $n$-th column of the adjacency matrix $A$, representing the connectivity of node $n$. |

A summary of Ogbn datasets' statistics in Table 2:

Table 2: Transposed Ogbn datasets statistics.

| Dataset | Ogbn-Arxiv | Dataset | Ogbn-Products |
|---|---|---|---|
| **Nodes** | 169,343 | **Nodes** | 2,449,029 |
| **Edges** | 1,166,243 | **Edges** | 61,859,140 |
| **Features** | 128 | **Features** | 100 |
| **Classes** | 40 | **Classes** | 47 |
| **Metric** | Accuracy | **Metric** | Accuracy |

**Lemma A.1.** *If $M \in \mathbb{R}^{n \times m}$ is a random matrix where $M_{i,j}$ are i.i.d. from $\mathcal{N}(0,1)$. Then,*

- *For any $t \geq 1$, with probability $1 - e^{-t^2}$, it satisfies*

$$\|M\|_2 \leq O\left(\sqrt{n} + \sqrt{m}\right) + t.$$

- *If $1 \leq s \leq O\left(\frac{m}{\log^2 m}\right)$, then with probability $1 - e^{-(n + s \log^2 m)}$ it satisfies*

$$\|Mv\|_2 \leq O\left(\sqrt{n} + \sqrt{s \log m}\right) \|v\|_2$$

*for all $s$-sparse vectors $v \in \mathbb{R}^m$.*

Proof: The statement can be found in Proposition B.2. from Allen-Zhu & Li (2019)

**Lemma A.2.** *Suppose $\delta \in [0,1]$ and $g^{(0)} \in \mathbb{R}^m$ is a random vector $g^{(0)} \sim \mathcal{N}(0, I_m)$. With probability at least $1 - e^{-\Omega(m\delta^{2/3})}$, for all vectors $g' \in \mathbb{R}^m$ with $\|g'\|_2 \leq \delta$, letting $D' \in \mathbb{R}^{m \times m}$ be the diagonal matrix where $(D')_{k,k} = \mathbf{1}_{(g^{(0)} + g')_k} - \mathbf{1}_{(g^{(0)})_k}$ for each $k \in [m]$, we have*

$$\|D'\|_0 \leq O(m^{2/3}) \quad and \quad \|D' g^{(0)}\|_2 \leq \|g'\|_2.$$

Proof: The statement can be found in Proposition B.4. from Allen-Zhu & Li (2019)

**Lemma A.3.** *Given a sampling set $X = \{x_n\}_{n=1}^N$ that contains $N$ partly dependent random variables, for each $n \in [N]$, suppose $x_n$ is dependent with at most $d_X$ random variables in $X$ (including $x_n$ itself), and the moment generating function of $x_n$ satisfies $\mathbb{E}[e^{sx_n}] \leq e^{Cs^2}$ for some constant $C$ that may depend on $x_n$. Then, the moment generation function of $\sum_{n=1}^N x_n$ is bounded as*

$$\mathbb{E}[e^{s\sum_{n=1}^N x_n}] \leq e^{Cd_X N s^2}.$$

Proof: The statement can be found in Lemma 7 from Zhang et al. (2020)

**Lemma A.4.** $\|Xa_n\| \leq \|A\|_1$.

Proof:

$$
\begin{aligned}
\|Xa_n\| &= \|\sum_{k=1}^N x_k a_{k,n}\| \\
&= \|\sum_{k=1}^N \frac{a_{k,n}}{\sum_{k=1}^N a_{k,n}} x_k\| \cdot \sum_{k=1}^N a_{k,n} \\
&\leq \sum_{k=1}^N \frac{a_{k,n}}{\sum_{k=1}^N a_{k,n}} \|x_k\| \cdot \|A\|_1 \\
&= \|A\|_1
\end{aligned}
\tag{21}
$$

**Lemma A.5.** *If $\mathcal{F} : \mathbb{R}^d \to \mathbb{R}^k$ has general complexity $(p, \mathcal{C}_s(\mathcal{F}), \mathcal{C}_\varepsilon(\mathcal{F}))$, then for every $x, y \in \mathbb{R}^d$, it satisfies $\|\mathcal{F}(x)\|_2 \leq \sqrt{k}p\mathcal{C}_s(\mathcal{F}) \cdot \|x\|_2$ and $\|\mathcal{F}(x) - \mathcal{F}(y)\|_2 \leq \sqrt{k}p\mathcal{C}_s(\mathcal{F}) \cdot \|x - y\|_2$.*

Proof: The boundedness of $\|\mathcal{F}(x)\|_2$ is trivial so we only focus on $\|\mathcal{F}(x) - \mathcal{F}(y)\|_2$. For each component $g(x) = \mathcal{F}_{r,i}\left(\frac{\langle w_{1,i}^*, (x,1)\rangle}{\|(x,1)\|_2}\right) \cdot \langle w_{2,i}^*, (x,1)\rangle$, denoting by $w_1^*$ as the first d coordinate of $w_{1,i}^*$, and by $w_{2,i}^*$ as the first d coordinates of $w_{2,i}^*$, we have

$$
\begin{aligned}
g'(x) &= \mathcal{F}_{r,i}\left(\frac{\langle w_{1,i}^*, (x,1)\rangle}{\|(x,1)\|_2}\right) \cdot w_2^* \\
&\quad + \langle w_{2,i}^*, (x,1)\rangle \cdot \mathcal{F}'_{r,i}\left(\frac{\langle w_{1,i}^*, (x,1)\rangle}{\|(x,1)\|_2}\right) \cdot \frac{w_1^* \cdot \|(x,1)\|_2 - \langle w_{1,i}^*, (x,1)\rangle \cdot (x,1)/\|(x,1)\|_2^2}{\|(x,1)\|_2^2}
\end{aligned}
$$

This implies

$$\|g'(x)\|_2 \leq \left|\mathcal{F}_{r,i}\left(\frac{\langle w_{1,i}^*, (x,1)\rangle}{\|(x,1)\|_2}\right)\right| + 2\left|\mathcal{F}'_{r,i}\left(\frac{\langle w_{1,i}^*, (x,1)\rangle}{\|(x,1)\|_2}\right)\right| \leq 3\mathcal{C}_s\left(\mathcal{F}_{r,i}\right)$$

As a result, $|\mathcal{F}_r(x) - \mathcal{F}_r(y)| \leq 3p\mathcal{C}_s\left(\mathcal{F}_{r,i}\right)$.

**Lemma A.6.** *For every smooth function $\phi$, every $\epsilon \in (0, \frac{1}{\mathcal{C}(\phi,a)\sqrt{a^2+1}})$, there exists a function $h : \mathbb{R}^2 \to [-\mathcal{C}_m(\phi,a)\sqrt{a^2+1}, \mathcal{C}_m(\phi,a)\sqrt{a^2+1}]$ that is also $\mathcal{C}_m(\phi,a)\sqrt{a^2+1}$-Lipschitz continuous on its first coordinate with the following two (equivalent) properties:*
*(a) For every $x_1 \in [-a, a]$ where $a > 0$:*

$$\left|\mathbb{E}\left[\mathbf{1}_{\alpha_1 x_1 + \beta_1\sqrt{a^2 - x_1^2} + b_0 \geq 0} h(\alpha_1, b_0)\right] - \phi(x_1)\right| \leq \epsilon$$

*where $\alpha_1, \beta_1, b_0 \sim \mathcal{N}(0,1)$ are independent random variables.*
*(b) For every $\mathbf{w}^*, \mathbf{x} \in \mathbb{R}^d$ with $\|\mathbf{w}^*\|_2 = 1$ and $\|\mathbf{x}\| \leq a$:*

$$\left|\mathbb{E}\left[\mathbf{1}_{\mathbf{w}\mathbf{X} + b_0 \geq 0} h(\mathbf{w}^\top \mathbf{w}^*, b_0)\right] - \phi(\mathbf{w}^{*\top}\mathbf{x})\right| \leq \epsilon$$

*where* $\mathbf{w} \sim \mathcal{N}(0, \mathbf{I})$ *is an d-dimensional Gaussian,* $b_0 \sim \mathcal{N}(0, 1)$.
*Furthermore, we have* $\mathbb{E}_{\alpha_1, b_0 \sim \mathcal{N}(0,1)}[h(\alpha_1, b_0)^2] \leq (\mathcal{C}_s(\phi, a))^2 (a^2 + 1)$.
*(c) For every* $\mathbf{w}^*, \mathbf{x} \in \mathbb{R}^d$ *with* $\|\mathbf{w}^*\|_2 = 1$, *let* $\tilde{\mathbf{w}} = (\mathbf{w}, b_0) \in \mathbb{R}^{d+1}$, $\tilde{\mathbf{x}} = (\mathbf{x}, 1) \in \mathbb{R}^{d+1}$ *with* $\|\tilde{\mathbf{x}}\| \leq \sqrt{a^2 + 1}$,
*then we have*

$$\left| \mathbb{E}\left[ \mathbf{1}_{\tilde{\mathbf{w}}^\top \tilde{\mathbf{x}} \geq 0} h(\tilde{\mathbf{w}}[1:d]^\top \mathbf{w}^*, \tilde{\mathbf{w}}[d+1]) \right] - \phi(\mathbf{w}^{*\top} \tilde{\mathbf{x}}[1:d]) \right| \leq \epsilon$$

*where* $\tilde{\mathbf{w}} \sim \mathcal{N}(0, \mathbf{I}_{d+1})$ *is an d-dimensional Gaussian.*
*We also have* $\mathbb{E}_{\tilde{\mathbf{w}} \in \mathcal{N}(0, \mathbf{I}_{d+1})}[h(\tilde{\mathbf{w}}[1:d]^\top \mathbf{w}^*, \tilde{\mathbf{w}}[d+1])^2] \leq (\mathcal{C}_s(\phi, a))^2 (a^2 + 1)$.

Proof: The statement can be found in Lemma B.1. from Li et al. (2022a)

## B  Concept Class

To quantify complexity in (8), we define model complexity $\mathcal{C}_m$ and sample complexity $\mathcal{C}_s$ as in Li et al. (2022a) (Section 1.2) and Allen-Zhu & Li (2019) (Section 4). For a smooth function $\phi(z) = \sum_{i=0}^{\infty} c_i z^i$:

$$\mathcal{C}_m(\phi, error, R) = \sum_{i=0}^{\infty} \left( (C^* R)^i + \left( \frac{\sqrt{\log(1/error)}}{\sqrt{i}} C^* R \right)^i \right) |c_i|, \tag{22}$$

$$\mathcal{C}_s(\phi, R) = C^* \sum_{i=0}^{\infty} (i+1)^{1.75} R^i |c_i|, \tag{23}$$

where $R \geq 0$ and $C^*$ is a sufficiently large constant. These two quantities are used in the model complexity and sample complexity, which represent the required number of model parameters and training samples to learn $\phi$ up to $error$, respectively. Many population functions have bounded complexity. For instance, if $\phi(z)$ is $\exp(z)$, $\sin(z)$, $\cos(z)$ or polynomials of $z$, then $\mathcal{C}_m(\phi, O(1)) \leq O(\text{poly}(1/error))$ and $\mathcal{C}_s(\phi, O(1)) \leq O(1)$.

Thus, the complexities of $\mathcal{F}$ and $\mathcal{G}$ are given by the tuples $(p_{\mathcal{F}}, \mathcal{C}_s(\mathcal{F}), \mathcal{C}_m(\mathcal{F}, error))$ and $(p_{\mathcal{G}}, \mathcal{C}_s(\mathcal{G}), \mathcal{C}_m(\mathcal{G}, error))$. $\mathcal{F}$ and $\mathcal{G}$ are composed by $p_{\mathcal{F}}$ and $p_{\mathcal{G}}$ different smooth functions.

We also state some simple properties regarding our complexity measure. We define $\mathcal{B}_{\mathcal{F}} := \max_n \|\mathcal{F}_{n,A^*}(X)\|_2$, $\mathcal{B}_{\mathcal{F} \circ \mathcal{G}} := \max_n \|\mathcal{G}_{n,A^*}(\mathcal{F}_{A^*}(X))\|_2$ for all $X$ satisfying $\|x_n\| = 1$. Assume $\mathcal{G}(\cdot)$ is $\mathcal{L}_{\mathcal{G}}$ Lipschitz continuous. It is simple to verify (see Lemma A.5) that $\mathcal{B}_{\mathcal{F}} \leq \sqrt{k} p_{\mathcal{F}} \mathcal{C}_s(\mathcal{F}, \|A^*\|_1) \|A^*\|_1$, $\mathcal{L}_{\mathcal{G}} \leq \sqrt{k} p_{\mathcal{G}} \mathcal{C}_s(\mathcal{G}, \mathcal{B}_{\mathcal{F}} \|A^*\|_1)$ and $\mathcal{B}_{\mathcal{F} \circ \mathcal{G}} \leq k p_{\mathcal{F}} \mathcal{C}_s(\mathcal{F}, \|A^*\|_1) \cdot p_{\mathcal{G}} \mathcal{C}_s(\mathcal{G}, \mathcal{B}_{\mathcal{F}} \|A^*\|_1) \mathcal{B}_{\mathcal{F}} \|A^*\|_1$.

## C  Theorem 3.2 Proof Details

Let us define learner networks that are single-skip two-hidden-layer with ReLU activation $\text{out}_n : \mathbb{R}^{d \times N} \times \mathbb{R}^N \to \mathbb{R}^k$ with

$$\text{out}_n(X, A; W, V) = \text{out}_n^1(X, A) + C\sigma(V \text{out}^1(X, A) a_n) \tag{24}$$

and

$$\text{out}^1(X, A) = C D_W \odot (W X A^1) \in R^{k \times N}, \tag{25}$$

$$\text{out}_n^1(X, A) = C D_W^n W X a_n^1 \in R^k \tag{26}$$

where

- $A = \begin{bmatrix} a_1, & a_2, & \cdots, & a_N \end{bmatrix} \in R^{N \times N}$ denotes the normalized adjacency matrix.

- $X \in R^{d \times N}$ denotes the the matrix of $d$ dimension features of $N$ nodes.

- $W \in R^{m \times d}$ denotes the first hidden layer weight.

- $V \in R^{m \times k}$ denotes the second hidden layer weight.

- $C \in R^{k \times m}$ denotes the output layer weight.

- $D_W = \begin{bmatrix} \mathbf{1}_{(WXa_1 \geq 0)}, & \mathbf{1}_{(WXa_2 \geq 0)}, & \cdots & , \mathbf{1}_{(WXa_N \geq 0)} \end{bmatrix}$, $D_W^n = \mathrm{diag}\{\mathbf{1}_{(WXa_n \geq 0)}\}$

- $D_V = \begin{bmatrix} \mathbf{1}_{(V\,\mathrm{out}_n^1(X,A)a_1 \geq 0)} & \mathbf{1}_{(V\,\mathrm{out}_n^1(X,A)a_2 \geq 0)} & \cdots & \mathbf{1}_{(V\,\mathrm{out}_n^1(X,A)a_N \geq 0)} \end{bmatrix}$, $D_V^n = \mathrm{diag}\{\mathbf{1}_{(V\,\mathrm{out}_n^1(X,A)a_n \geq 0)}\}$

The $l_2$ loss is represented as a function of the weight deviations $\mathbf{W}, \mathbf{V}$ from initiation $W^{(0)}$ and $V^{(0)}$, i.e.,

$$L(\mathbf{W}, \mathbf{V}) = \mathrm{Obj}_n(X, A^{1t}, A^{2t}, y_n; W^{(0)} + \mathbf{W}, V^{(0)} + \mathbf{V}). \tag{27}$$

Let $W^{(t)} = W^{(0)} + \mathbf{W}_t$, $V^{(t)} = V^{(0)} + \mathbf{V}_t$. We assume $0 < \alpha \leq \widetilde{O}\left(\frac{1}{kp_{\mathcal{G}}\mathcal{C}_s(\mathcal{G}, \mathcal{B}_{\mathcal{F}}\|A^*\|_1)}\right)$ throught the training. We prove that $\|\mathbf{W}_t\|_2$ and $\|\mathbf{V}_t\|_2$ are bounded by $\tau_w$ and $\tau_v$ in Table 3 during training, i.e., $\|\mathbf{W}_t\|_2 \leq \tau_w$, $\|\mathbf{V}_t\|_2 \leq \tau_v$ for all $t$. See Appendix C.4 for the proof.

Table 3: Parameter choices

| | |
|---|---|
| $\sigma_w$ | $m^{-\frac{1}{2}+0.01} \leq \sigma_w \leq m^{-0.01}$    $\sigma_v$    $\sigma_v = \Theta(\mathrm{polylog}(m))$ |
| $\tau_w$ | $\tau_w = \widetilde{\Theta}\left(kp_{\mathcal{F}}\mathcal{C}_s(\mathcal{F}, \|A^*\|_1)\right)$ and $m^{\frac{1}{8}+0.001}\sigma_w \leq \tau_w \leq m^{\frac{1}{8}-0.001}\sigma_w^{\frac{1}{4}}$ |
| $\tau_v$ | $\tau_v = \widetilde{\Theta}\left(\alpha kp_{\mathcal{G}}\mathcal{C}_s(\mathcal{G}, \mathcal{B}_{\mathcal{F}}\|A^*\|_1)\mathcal{B}_{\mathcal{F}}\|A^*\|_1\right)$ and $\sigma_v\left(\frac{k}{m}\right)^{\frac{3}{8}} \leq \tau_v \leq \frac{\sigma_v}{\mathrm{polylog}(m)\|A^*\|_1} < 1$ |

## C.1 Coupling

**Lemma C.1.** *We show that the weights after a properly bounded amount of updates stay close to the initialization, and many good properties occur. Suppose that $\|\mathbf{W}\|_2 \leq \tau_w$, $\|\mathbf{V}\|_2 \leq \tau_v$, $W_0$ from $\mathcal{N}\left(0, \sigma_w^2\right)$ and $V_0$ from $\mathcal{N}\left(0, \sigma_v^2/m\right)$, we have that*

*1.*

$$\left\|D_{\mathbf{W}^{(0)}}^n - D_{\mathbf{W}+\mathbf{W_0}}^n\right\|_0 \leq O\left((\frac{\tau_w}{\sigma_w})^{2/3}m^{2/3}\right) \tag{28}$$

*2.*

$$\left\|\mathbf{C}D_{\mathbf{W}+\mathbf{W_0}}^n \mathbf{W}Xa_n - \mathbf{C}D_{\mathbf{W}+\mathbf{W_0}}^n\left(\mathbf{W}+\mathbf{W_0}\right)Xa_n\right\|_2 \leq \widetilde{O}\left(\frac{\sqrt{s}}{\sqrt{m}}\tau_w\|A\|_1 + \sqrt{k}\sigma_w\|A\|_1\right) \tag{29}$$

*3.*

$$\left\|\mathrm{out}_n^1(X, A; W)\right\|_2 \leq \widetilde{O}(\tau_w\|A\|_1) \tag{30}$$

*4.*

$$\left\|D_{\mathbf{V}^{(0)}}^n - D_{\mathbf{V}+\mathbf{V_0}}^n\right\|_0 \leq O\left((\frac{\tau_v}{\sigma_v})^{2/3}m\right) \tag{31}$$

*5.*

$$\left\|\mathbf{C}D_{\mathbf{V}+\mathbf{V_0}}^n \mathbf{V}\,\mathrm{out}^1(X, A)a_n - \mathbf{C}D_{\mathbf{V}+\mathbf{V_0}}^n\left(\mathbf{V}+\mathbf{V_0}\right)\mathrm{out}^1(X, A)a_n\right\|_2$$
$$\leq \widetilde{O}\left((\frac{\sqrt{k}}{\sqrt{m}}\sigma_v + \frac{\sqrt{s}}{\sqrt{m}}\tau_v)\|\mathrm{out}_n^1(X, A)\|_2\|A\|_1\right) \tag{32}$$

*6.*

$$\left\|\mathbf{C}D_{\mathbf{V}+\mathbf{V_0}}^n \mathbf{V_0}\right\|_2 \leq \tau_v(\frac{\tau_v}{\sigma_v})^{1/3} \tag{33}$$

7.
$$\|\mathbf{C}D^n_{\mathbf{V}+\mathbf{V_0}}\left(\mathbf{V}+\mathbf{V_0}\right)\mathrm{out}^1(X,A)a_n\|_2 \leq \widetilde{O}\left(\tau_v\|\mathrm{out}^1_n(X,A)\|_2\|A\|_1\right) \tag{34}$$

Proof:

1. $\|WXa_n\|_2 \leq \|W\|_2\|Xa_n\|_2 \leq \tau_w\|Xa_n\|_2$ and $\langle W_0, Xa_n\rangle_j \sim \mathcal{N}(0, \|Xa_n\|_2^2\sigma_w^2)$, using Lemma A.2, we have
$$\left\|D^n_{\mathbf{W}^{(0)}} - D^n_{\mathbf{W}+\mathbf{w_o}}\right\|_0 \leq O\left((\frac{\tau_w\|Xa_n\|_2}{\sigma_w\|Xa_n\|_2\sqrt{m}})^{2/3}m\right) \tag{35}$$

2. We write $\mathbf{C}D^n_{\mathbf{W}+\mathbf{W_o}}\mathbf{W}Xa_n^{*1} - \mathbf{C}D^n_{\mathbf{W}+\mathbf{W_o}}\left(\mathbf{W}+\mathbf{W}_0\right)Xa_n^{*1} = -\mathbf{C}D^n_{\mathbf{W_0}}\mathbf{W}^{(0)}Xa_n^{*1} + \mathbf{C}\left(D^n_{\mathbf{W_0}} - D^n_{\mathbf{W}+W_0}\right)\mathbf{W}_0Xa_n^{*1}$. For the first term, $\left\|D^n_{\mathbf{W_0}}\mathbf{W}_0Xa_n\right\|_2 \leq \|\mathbf{W}_0Xa_n\|_2 \leq O\left(\sigma_w\|A\|_1\sqrt{m}\right)$, so $\left\|CD^n_{\mathbf{W_0}}\mathbf{W}_0Xa_n\right\|_2 \leq \widetilde{O}\left(\sqrt{k}\sigma_w\|A\|_1\right)$

    For the second term, using Lemma A.2 again, we have
    $$\left\|\left(D^n_{\mathbf{W_0}} - D^n_{\mathbf{W}+W_0}\right)\mathbf{W}_0Xa_n\right\|_2 \leq \|WXa_n\|_2 \leq \tau_w\|A\|_1$$

    Using Lemma A.1, for every $s$-sparse vector $\mathbf{y}$, it satisfies $\|\mathbf{A}\mathbf{y}\|_2 \leq e^{O(\sqrt{\frac{s}{m}})}\|\mathbf{y}\|_2$ with high probability. The sparsity of the second term is $s = (\frac{\tau_w}{\sigma_w\|A\|_1})^{2/3}m^{2/3}$, so we have $\left\|\mathbf{C}\left(D^n_{\mathbf{W_0}} - D^n_{\mathbf{W}+W_0}\right)\mathbf{W}_0Xa_n\right\|_2 \leq \widetilde{O}\left(\frac{\sqrt{s}}{\sqrt{m}}\right)\cdot\|\mathbf{W}Xa_n\|_2 \leq \widetilde{O}\left(\frac{\sqrt{s}}{\sqrt{m}}\tau_w\|A\|_1\right)$.

3. $\left\|\mathrm{out}^1_n(X,A)\right\|_2 \leq \left\|\mathbf{C}D^n_{\mathbf{W}+\mathbf{W_o}}\mathbf{W}Xa_n\right\|_2 + \left\|\mathbf{C}D^n_{\mathbf{W}+\mathbf{W_o}}\mathbf{W}Xa_n - \mathbf{C}D^n_{\mathbf{W}+\mathbf{W_o}}\left(\mathbf{W}+\mathbf{W}_0\right)Xa_n\right\|_2$. Using $\|C\|_2 \leq 1$ with high probability, we have $\left\|\mathbf{C}D^n_{\mathbf{W}+\mathbf{W_o}}Xa_n\right\|_2 \leq \widetilde{O}(\tau_w\|A\|_1)$

4. Similar to (28), $\left\|V\mathrm{out}^1(X,A)a_n\right\|_2 \leq \tau_v\left\|\mathrm{out}^1(X,A)a_n\right\|_2$ and $\left\langle V_0, \mathrm{out}^1(X,A)a_n\right\rangle_j \sim \mathcal{N}(0, \left\|\mathrm{out}^1(X,A)a_n\right\|_2^2\sigma_v^2)$, using Lemma A.2 we can prove it.

5. We write $\mathbf{C}D^n_{\mathbf{V}+\mathbf{V_o}}\mathbf{V}\mathrm{out}^1(X,A)a_n - \mathbf{C}D^n_{\mathbf{V}+\mathbf{V_o}}\left(\mathbf{V}+\mathbf{V}_0\right)\mathrm{out}^1(X,A)a_n = -\mathbf{C}D^n_{\mathbf{V_0}}\mathbf{V}^{(0)}\mathrm{out}^1(X,A)a_n + \mathbf{C}\left(D^n_{\mathbf{V_0}} - D^n_{\mathbf{V}+V_0}\right)\mathbf{V}_0\mathrm{out}^1(X,A)a_n$. Similar to (29), we have $\|\mathbf{C}D^n_{\mathbf{V_0}}\mathbf{V}^{(0)}\mathrm{out}^1(X,A)a_n\|_2 \leq \widetilde{O}(\sqrt{k}/\sqrt{m})\cdot O\left(\sigma_v\|\mathrm{out}^1(X,A)a_n\|_2\right)$ and $\left\|\mathbf{C}\left(D^n_{\mathbf{V_0}} - D^n_{\mathbf{V}+V_0}\right)\mathbf{V}_0\mathrm{out}^1(X,A)a_n\right\|_2 \leq \widetilde{O}\left(\frac{\sqrt{s}}{\sqrt{m}}\tau_v\|\mathrm{out}^1(X,A)a_n\|_2\right)$. $\|\mathrm{out}^1(X,A)a_n\|_2 \leq \|\mathrm{out}^1_n(X,A)\|_2\|A\|_1$

6. From 5, it is easy to get.

7. From 3, it is easy to get.

## C.2 Existantial

Consider random function $S_n\left((X,A); \mathbf{W}^*\right) = \left(S^1_n\left((X,A); \mathbf{W}^*\right), \dots, S^k_n\left((X,A); \mathbf{W}^*\right)\right)$ in which

$$S^r_n\left((X,A); \mathbf{W}^*\right) \stackrel{\mathrm{def}}{=} \sum_{i=1}^m a_{r,i}\cdot\langle w^*_i, Xa_n\rangle\cdot\mathbf{1}_{\left\langle w^{(0)}_i, Xa_n\right\rangle\geq 0} \tag{36}$$

where $W^*$ is a given matrix, $W^0$ is a random matrix where each $w^{(0)}_i$ is i.i.d. from $\mathcal{N}\left(0, \frac{\mathbf{I}}{m}\right)$ and $a_{r,i}$ is i.i.d. from $\mathcal{N}(0,1)$.

Based on Lemma B.1. from Li et al. (2022a) and Lemma E.1. from **?**, we have

**Lemma C.2.** *Given any* $\mathcal{F}: \mathbb{R}^d \to \mathbb{R}^k$ *with general complexity* $(p, \mathcal{C}_s(\mathcal{F}, \|A\|_1)\|A\|_1, \mathcal{C}_\varepsilon(\mathcal{F}, \|A\|_1)\|A\|_1)$, *for every* $\epsilon \in \left(0, \frac{1}{pk\mathcal{C}_s(\mathcal{F}, \|A\|_1)\|A\|_1}\right)$, *there exist* $M = \mathrm{poly}\left(\mathcal{C}_\varepsilon(\mathcal{F}, \|A\|_1), \|A\|_1, 1/\varepsilon\right)$ *such that if* $m \geq M$, *then with high probability there is a construction* $\mathbf{W}^* = (w^*_1, \dots, w^*_m) \in \mathbb{R}^{m\times d}$ *with*

$$\|\mathbf{W}^*\|_{2,\infty} \leq \frac{kp\mathcal{C}_\varepsilon(\mathcal{F}, \|A\|_1), \|A\|_1}{m} \quad and \quad \|\mathbf{W}^*\|_F \leq \widetilde{O}\left(\frac{kp\mathcal{C}_s(\mathcal{F}, \|A\|_1)\|A\|_1}{\sqrt{m}}\right) \tag{37}$$

*satisfying, for every $x_n \in R^d$ and $\|x_n\|_2 \leq 1$, with probability at least $1 - e^{-\Omega(\sqrt{m})}$*

$$\sum_{r=1}^{k} |\mathcal{F}_n^r(X, A) - S_n^r(X, A; W^*)| \leq \varepsilon \cdot \|A\|_1 \tag{38}$$

*where $G_n(X, A; W^*) = \begin{bmatrix} S_n^1(X, A; W^*) \\ \vdots \\ S_n^k(X, A; W^*) \end{bmatrix}$ and $S_n(X, A; W^*) = CD_{W+W_0}^n W^* X a_n$*

Proof: Define $w_j^* = \sum_{r\in[k]} a_{r,j} \sum_{i\in[p]} a_{r,i}^* h^{(r,i)} \left( \sqrt{m} \left\langle w_j^{(0)}, w_{1,i}^* \right\rangle \right) w_{2,i}^*$ has the same distribution with $\alpha_1$ in Lemma A.6.

Using Lemma A.6 we have $\left| h^{(r,i)} \right| \leq \mathcal{C}_\varepsilon(\mathcal{F}, \|A\|_1) \|A\|_1$ and using Lemma E.1. from **?**, we have for our parameter choice of $m$, with probability at least $1 - e^{-\Omega\left(m\varepsilon^2 / \left(k^4 p^2 \mathcal{C}_\varepsilon(\mathcal{F}, \|A\|_1)\|A\|_1\right)\right)}$

$$|\mathcal{F}_n^r(X, A) - S_n^r(X, A; W^*)| \leq \frac{\varepsilon}{k}.$$

We have for each $j \in [m]$, with high probability $\|w_j^*\|_2 \leq \widetilde{O}\left( \frac{kp\mathcal{C}_\varepsilon(\mathcal{F}, \|A\|_1)\|A\|_1}{m} \right)$. This means $\|\mathbf{W}^*\|_{2,\infty} \leq \widetilde{O}\left( \frac{kp\mathcal{C}_\varepsilon(\mathcal{F}, \|A\|_1)\|A\|_1}{m} \right)$. As for the Frobenius norm,

$$\|\mathbf{W}^*\|_F^2 = \sum_{j\in[m]} \|w_j^*\|_2^2 \leq \sum_{j\in[m]} \widetilde{O}\left( \frac{k^2 p}{m^2} \right) \cdot \sum_{i\in[p]} h^{(r,i)} \left( \sqrt{m} \left\langle w_j^{(0)}, w_{1,i}^* \right\rangle \right)^2 \tag{39}$$

Applying Hoeffding's concentration, we have with probability at least $1 - e^{-\Omega(\sqrt{m})}$

$$\begin{aligned} \sum_{j\in[m]} h^{(r,i)} \left( \sqrt{m} \left\langle w_j^{(0)}, w_{1,i}^* \right\rangle, \sqrt{m} b_j^{(0)} \right)^2 &\leq m \cdot (\mathcal{C}_s(\mathcal{F}, \|A\|_1)\|A\|_1^2), \\ &\quad + m^{3/4} \cdot (\mathcal{C}_\varepsilon(\mathcal{F}, \|A\|_1)\|A\|_1)^2, \\ &\leq 2m(\mathcal{C}_s(\mathcal{F}, \|A\|_1)\|A\|_1)^2. \end{aligned} \tag{40}$$

Putting this back to (39) we have $\|\mathbf{W}^*\|_F^2 \leq \widetilde{O}\left( \frac{k^2 p^2 (\mathcal{C}_s(\mathcal{F}, \|A\|_1)\|A\|_1)^2}{m} \right)$.

**Lemma C.3.** *Under the assumptions of Lemma C.1, suppose $\alpha \in (0,1)$ and $\widetilde{\alpha} = \frac{\alpha}{k(p_\mathcal{F}\mathcal{C}_s(\mathcal{F}, \|A\|_1) + p_\mathcal{G}\mathcal{C}_s(\mathcal{G}, \|A\|_1))}$, there exist $M = \mathrm{poly}\left( \mathcal{C}_{\widetilde{\alpha}}(\mathcal{F}, \|A\|_1), \mathcal{C}_{\widetilde{\alpha}}(\mathcal{G}, \|A\|_1), \|A\|_1, \widetilde{\alpha}^{-1} \right)$ satisfying that for every $m \geq M, \|\mathbf{W}^*\|_F \leq \widetilde{O}(kp_\mathcal{F}\mathcal{C}_s(\mathcal{F}))$ and $\|\mathbf{V}^*\|_F \leq \widetilde{O}(\widetilde{\alpha}kp_\mathcal{G}\mathcal{C}_s(\mathcal{G}))$ with high probability*

*1.*
$$\mathbb{E}_{n\in\mathcal{V},(X,y_n)\sim\mathcal{D}} \left[ \left\| \mathbf{C}D_{\mathbf{W_0}}^n \mathbf{W}^* X a_n - \mathcal{F}_n(X, A) \right\|_2 \right] \leq \widetilde{\alpha}^2 \|A\|_1 \tag{41}$$

*2.*
$$\left\| \mathbf{C}D_{\mathbf{V_0}}^n \mathbf{V}^* \mathrm{out}^1(X, A) a_n - \alpha \mathcal{G}_n \left( \mathrm{out}^1(X, A), A \right) \right\|_2 \leq \widetilde{\alpha}^2 \cdot \left\| \mathrm{out}_n^1(X, A) \right\|_2 \|A\|_1 \tag{42}$$

*3.*
$$\mathbb{E}_{n\in\mathcal{V},(X,y_n)\sim\mathcal{D}} \left[ \left\| \mathbf{C}D_{\mathbf{W}}^n \mathbf{W}^* X a_n - \mathcal{F}_n(X, A) \right\|_2 \right] \leq O(\widetilde{\alpha}^2 \|A\|_1) \tag{43}$$

*4.*
$$\begin{aligned} &\left\| \mathbf{C}D_{\mathbf{V}}^n \mathbf{V}^* \mathrm{out}^1(X, A) a_n - \alpha \mathcal{G}_n \left( \mathrm{out}^1(X, A), A \right) \right\|_2 \\ &\leq \left( \widetilde{\alpha}^2 + O\left( \tau_v \left( \frac{\tau_v}{\sigma_v} \right)^{1/3} \right) \right) \left\| \mathrm{out}_n^1(X, A) \right\|_2 \|A\|_1 \end{aligned} \tag{44}$$

5.

$$\mathbb{E}_{n\in\mathcal{V},(X,y_n)\sim\mathcal{D}}\left[\left\|\mathbf{C}D_{\mathbf{W_0}+\mathbf{W}}^n(\mathbf{W}^* - \mathbf{W})Xa_n - (\mathcal{F}_n(X,A) - \text{out}_n^1(X,A))\right\|_2\right] \leq \widetilde{\alpha}^2\left\|A\right\|_1 \qquad (45)$$

Proof:

1. Using Lemma C.2, we can find a $\mathbf{W}^*$ satisfying $\left\|\mathbf{C}D_{\mathbf{W_0}+\mathbf{W}}^n\mathbf{W}^*Xa_n - \mathcal{F}_n(X,A)\right\|_2$ small enough with probability at least $1 - e^{-\Omega(\sqrt{m})}$.

2. Using Lemma C.2 and $\left\|\text{out}^1(X,A)a_n\right\|_2 \leq \|\text{out}_n^1(X,A)\|_2\|A\|_1$, we can easily prove it.

3. $\|\mathbf{W}^*Xa_n\|_2 \leq O(\|\mathbf{W}^*\|_F\|Xa_n\|_2) \leq O(\tau_w\|A\|_1)$. $\left\|\mathbf{C}(D_{\mathbf{W}}^n - D_{\mathbf{W_0}}^n)\mathbf{W}^*Xa_n\right\|_2 \leq O(\sqrt{s}\tau_w\|A\|_1/\sqrt{m})$ where $s$ is the maximum sparsity of $(D_{\mathbf{W}}^n - D_{\mathbf{W_0}}^n)$. Using (28), we know $s \leq O\left((\frac{\tau_w}{\sigma_w})^{2/3}m^{2/3}\right)$. This, combining with (41) gives

$$\mathbb{E}_{n\in\mathcal{V},(X,y_n)\sim\mathcal{D}}\left[\|\mathbf{C}D_{\mathbf{W}}^n\mathbf{W}^*Xa_n - \mathcal{F}_n(X,A)\|_2\right] \leq \widetilde{\alpha}^2\|A\|_1 + O\left(\tau_w(\frac{\tau_w}{\sigma_w})^{1/3}/m^{1/6}\right)$$
$$\leq O(\widetilde{\alpha}^2\|A\|_1) \qquad (46)$$

4. Using (31) and $\left\|\mathbf{V}^*\text{out}^1(X,A)a_n\right\|_2 \leq O\left(\tau_v\|\text{out}_n^1(X,A)\|_2\|A\|_1\right)$ we can easily prove it.

5. Using (29) and (43), with larger enough $m$, we can prove it.

## C.3 Optimization

We write the gradient of loss function as $\nabla_{\mathbf{W}}\text{Obj}_n(\mathbf{W}) = \nabla_{\mathbf{W}}\text{Obj}_n^1(\mathbf{W}) + \nabla_{\mathbf{W}}\text{Obj}_n^2(\mathbf{W})$, where $\nabla_{\mathbf{W}}\text{Obj}_n^1(\mathbf{W}) = \nabla_{\mathbf{W}}\text{out}_n^1(X,A)$ and $\nabla_{\mathbf{W}}\text{Obj}_n^2(\mathbf{W}) = \nabla_{\mathbf{W}}CD_V^nV\text{out}^1(X,A)a_n$, we can write its gradient as follows.

$$\begin{aligned}\left\langle\nabla_{\mathbf{W}}\text{Obj}_n^1(\mathbf{W}), -\mathbf{W}'\right\rangle &= tr(Xa_n(y_n - \text{out}_n(X,A))^\top CD_{W+W_0}^n\mathbf{W}') \\ &= tr((y_n - \text{out}_n(X,A))^\top CD_{W+W_0}^nW'Xa_n) \\ &= \left\langle y_n - \text{out}_n(X,A), CD_{W+W_0}^nW'Xa_n\right\rangle\end{aligned} \qquad (47)$$

$$\begin{aligned}\left\langle\nabla_{\mathbf{W}}\text{Obj}_n^2(\mathbf{W}), -\mathbf{W}'\right\rangle &= tr(\sum_{i=1}^N a_{ni}Xa_i(y_n - \text{out}_n(X,A))^\top CD_{V+V_0}^n(\mathbf{V}^{(0)} + \mathbf{V})CD_{W+W_0}^i\mathbf{W}') \\ &= tr(\sum_{i=1}^N a_{ni}(y_n - \text{out}_n(X,A))^\top CD_{V+V_0}^n(\mathbf{V}^{(0)} + \mathbf{V})CD_{W+W_0}^i\mathbf{W}'Xa_i) \\ &= tr((y_n - \text{out}_n(X,A))^\top\sum_{i=1}^N a_{ni}CD_{V+V_0}^n(\mathbf{V}^{(0)} + \mathbf{V})CD_{W+W_0}^i\mathbf{W}'Xa_i) \\ &= \left\langle y_n - \text{out}_n(X,A), CD_{V+V_0}^n(\mathbf{V}^{(0)} + \mathbf{V})C(D_{W+W_0}\odot W'XA)a_n\right\rangle\end{aligned} \qquad (48)$$

$$\begin{aligned}\left\langle\nabla_{\mathbf{V}}\text{Obj}_n(\mathbf{V}), -\mathbf{V}'\right\rangle &= tr(\text{out}(X)a_n(y_n - \text{out}_n(X,A))^\top CD_{V+V_0}^n\mathbf{V}') \\ &= tr((y_n - \text{out}_n(X,A))^\top CD_{V+V_0}^n\mathbf{V}'\text{out}(X)a_n) \\ &= \left\langle y_n - \text{out}_n(X,A), CD_{V+V_0}^n\mathbf{V}'\text{out}(X)a_n\right\rangle\end{aligned} \qquad (49)$$

Let us define $f(\mathbf{W}') = CD_{W+W_0}^nW'Xa_n + CD_{V+V_0}^n(\mathbf{V}^{(0)} + \mathbf{V})C(D_{W+W_0}\odot W'XA)a_n$ and $g(\mathbf{V}') = CD_{V+V_0}^n\mathbf{V}'\text{out}(X)a_n$. Therefore,

$$\left\langle\nabla_{\mathbf{W},\mathbf{V}}\text{Obj}_n(\mathbf{W},\mathbf{V}), (-\mathbf{W}', -\mathbf{V}')\right\rangle = \left\langle y_n - \text{out}_n(X,A), f(\mathbf{W}') + g(\mathbf{V}')\right\rangle \qquad (50)$$

**Claim C.4.** *We have that for all $\mathbf{W}$ and $\mathbf{V}$ satisfying $\|\mathbf{W}\|_F \leq \tau_w$ and $\|\mathbf{V}\|_F \leq \tau_v$, it holds that*

$$\|\nabla_{\mathbf{W}} \text{Obj}(\mathbf{W}, \mathbf{V}; (x, y))\|_F \leq \|A\|_1 \|y_n - \text{out}_n(X, A)\|_2 \cdot O(\sigma_v + 1) \tag{51}$$

$$\|\nabla_{\mathbf{V}} \text{Obj}(\mathbf{W}, \mathbf{V}; (x, y))\|_F \leq \tau_w \|A\|_1 \|y_n - \text{out}_n(X, A)\|_2 \cdot O(1) \tag{52}$$

Proof:

$$\begin{aligned}
\|\nabla_{\mathbf{W}} \text{Obj}(\mathbf{W}, \mathbf{V}; (x, y))\|_F &= \|X a_n (y_n - \text{out}_n(X, A))^\top \\
&\quad \times (C D_{W+W_0}^n + C D_{V+V_0}^n (\mathbf{V}^{(0)} + \mathbf{V}) C D_{W+W_0}^n)\|_F \\
&\leq \|X a_n\|_2 \|y_n - \text{out}_n(X, A)\|_2 \\
&\quad \times \|C D_{W+W_0}^n + C D_{V+V_0}^n (\mathbf{V}^{(0)} + \mathbf{V}) C D_{W+W_0}^n\|_2 \\
&\leq \|A\|_1 \|y_n - \text{out}_n(X, A)\|_2 \cdot O(\sigma_v + 1)
\end{aligned} \tag{53}$$

In (53), the last inequality uses $\|V^{(0)}\|_2 = O(\tau_v)$ and $\|C\|_2 \leq 1$.

$$\begin{aligned}
\|\nabla_{\mathbf{V}} \text{Obj}(\mathbf{W}, \mathbf{V}; (x, y))\|_F &= \| \text{out}_n(X, A) a_n (y_n - \text{out}_n(X, A))^\top C D_{V+V_0}^n\|_F \\
&\leq \| \text{out}_n(X, A) a_n\|_2 \|y_n - \text{out}_n(X, A)\|_2 \|C D_{V+V_0}^n\|_2 \\
&\leq \tau_w \|A\|_1 \|y_n - \text{out}_n(X, A)\|_2 \cdot O(1)
\end{aligned} \tag{54}$$

In (54), the last inequality uses (30) and $\|C\|_2 \leq 1$.

**Claim C.5.** *In the setting of Lemma C.1, we have $f(\mathbf{W}^* - \mathbf{W}) + g(\mathbf{V}^* - \mathbf{V}) = \mathcal{H}_{n,A^*}(X, A) - \text{out}_n(X, A) + Err_n$ with*

$$\begin{aligned}
\mathbb{E}_{n \in \mathcal{V}, (X, y_n) \sim \mathcal{D}} \|Err_n\|_2 &\leq \mathbb{E}_{n \in \mathcal{V}, (X, y_n) \sim \mathcal{D}} [O(\tau_v \|A\|_1 + \alpha \mathcal{L}_{\mathcal{G}} \|A\|_1) \cdot \|\mathcal{H}_{n,A^*}(X, A) - \text{out}_n(X, A)\|_2] \\
&\quad + O\left(\tau_v^2 \|A\|_1^2 \mathcal{B}_{\mathcal{F}} + \tau_v \widetilde{\alpha}^2 \|A\|_1^2 + \alpha \tau_v \|A\|_1^2 \mathcal{L}_{\mathcal{G}} \mathcal{B}_{\mathcal{F}}\right)
\end{aligned} \tag{55}$$

Proof: Based on the definition of $f(\mathbf{W}')$ and $g(\mathbf{V}')$, we have

$$\begin{aligned}
f(\mathbf{W}^* - \mathbf{W}; X, a_n^*) + g(\mathbf{V}^* - \mathbf{V}; X, a_n^*) &= C D_{W+W_0}^n (W^* - W) X a_n \\
&\quad + C D_{V+V_0}^n (V^* - V) \text{out}(X) a_n \\
&\quad + C D_{V+V_0}^n (\mathbf{V}^{(0)} + \mathbf{V}) C (D_{W+W_0} \odot (W^* - W) X A) a_n \\
&= \underbrace{C D_{V+V_0}^n (\mathbf{V}^{(0)} + \mathbf{V}) C (D_{W+W_0} \odot (W^* - W) X A) a_n}_{\clubsuit} \\
&\quad + \underbrace{C D_{W+W_0}^n W^* X a_n + C D_{V+V_0}^n V^* \text{out}^1(X, A) a_n}_{\spadesuit} \\
&\quad - \underbrace{(C D_{W+W_0}^n W X a_n + C D_{V+V_0}^n V \text{out}^1(X, A) a_n)}_{\diamondsuit}
\end{aligned} \tag{56}$$

1. For the $\clubsuit$ term,

$$\begin{aligned}
\clubsuit &\leq \left(\left\|C D_{V+V_0}^n \mathbf{V}^{(0)}\right\|_2 + \|\mathbf{C}\|_2^2 \|\mathbf{V}\|_2\right) \|C(D_{W+W_0} \odot (W^* - W) X A) a_n\|_2 \\
&\leq O(1) \cdot O(\tau_v) \cdot \sum_{i=1}^N a_{ni} \left(\|\mathcal{F}(x) - \text{out}_n^1(X, A)\|_2 + O(\widetilde{\alpha}^2 \|A\|_1)\right) \\
&\leq O(\tau_v) \left(\|\mathcal{F}(x) - \text{out}_i(x)\|_2 \|A\|_1 + O(\widetilde{\alpha}^2 \|A\|_1^2)\right)
\end{aligned} \tag{57}$$

together with $\tau_v \leq \frac{1}{\text{polylog}(m)} \sigma_v$.

2. For the ♠ term,

$$
\begin{aligned}
\spadesuit - (\mathcal{F}_n(X,A) + \alpha\mathcal{G}(\mathcal{F}(x), a_n) &= C D_{W+W_0}^n W^* X a_n - \mathcal{F}_n(X,A) \\
&\quad + C D_{V+V_0}^n V^* \operatorname{out}_n^1(X,A) a_n - \alpha \mathcal{G}\left(\operatorname{out}_n^1(X,A), a_n\right) \\
&\quad + \alpha \mathcal{G}\left(\operatorname{out}_n^1(X,A), a_n\right) - \alpha \mathcal{G}\left(\mathcal{F}(x), a_n\right)
\end{aligned}
\tag{58}
$$

The first term uses (41), the second term uses (42) and the third term uses the Lipscthiz continuity of $\mathcal{G}$, so we have

$$
\begin{aligned}
\left\| \spadesuit - (\mathcal{F}(x) + \alpha\mathcal{G}(\mathcal{F}(x)))\right\|_2 &\le O\left(\widetilde{\alpha}^2 + \tau_v (\tfrac{\tau_v}{\sigma_v})^{1/3}\right) \cdot \left\|\operatorname{out}_n^1(X, a_n)\right\|_2 \|A\|_1 \\
&\quad + O\left(\alpha\mathcal{L}_\mathcal{G}\right) \left\|\mathcal{F}(X) a_n - \operatorname{out}_n^1(X) a_n\right\|_2 \\
&\le O\left(\tau_v^2\right) \cdot \left\|\operatorname{out}_n^1(x)\right\|_2 \|A\|_1 + O\left(\alpha\mathcal{L}_\mathcal{G}\right) \left\|\mathcal{F}_n(x) - \operatorname{out}_n^1(x)\right\|_2 \|A\|_1
\end{aligned}
\tag{59}
$$

We use $\frac{1}{\sigma_v} \le \tau_v^2$ and definition of $\widetilde{\alpha}$.

3. For the ♢ term,

$$
\left\| \diamondsuit - \operatorname{out}_n(X)\right\|_2 \le O\left(\left(\left\|\operatorname{out}_n^1(x)\right\|_2 \|A\|_1\right) \tau_v^2\right)
\tag{60}
$$

where the inequality uses (29) and (32).

In sum, we have

$$
Err \overset{\text{def}}{=} f\left(\mathbf{W}^* - \mathbf{W}; x\right) + g\left(\mathbf{V}^* - \mathbf{V}; x\right) - (\mathcal{F}(x) + \alpha\mathcal{G}(\mathcal{F}(x)) - \operatorname{out}_n(X,A)
\tag{61}
$$

satisfy

$$
\begin{aligned}
\mathop{\mathbb{E}}_{n\in\mathcal{V},(X,y_n)\sim\mathcal{D}} \|Err\|_2 &\le \mathop{\mathbb{E}}_{n\in\mathcal{V},(X,y_n)\sim\mathcal{D}} \Bigg[ O\left(\tau_v \|A\|_1 + \alpha\mathcal{L}_\mathcal{G}\|A\|_1\right) \\
&\quad\quad \times \left\|\mathcal{F}(x) - \operatorname{out}_n^1(X,A)\right\|_2 + O\left(\left\|\operatorname{out}_n^1(x)\right\|_2 \|A\|_1\right) \tau_v^2 \Bigg] \\
&\quad + O\left(\tau_v \widetilde{\alpha}^2 \|A\|_1^2\right)
\end{aligned}
\tag{62}
$$

Using $\left\|\operatorname{out}_n^1(x)\right\|_2 \le \left\|\operatorname{out}_n^1(X,A) - \mathcal{F}(x)\right\|_2 + \mathcal{B}_\mathcal{F}$, we have

$$
\begin{aligned}
\mathop{\mathbb{E}}_{n\in\mathcal{V},(X,y_n)\sim\mathcal{D}} \|Err\|_2 &\le \mathop{\mathbb{E}}_{n\in\mathcal{V},(X,y_n)\sim\mathcal{D}} \left[O\left(\tau_v \|A\|_1 + \alpha\mathcal{L}_\mathcal{G}\|A\|_1\right) \cdot \left\|\mathcal{H}(x) - \operatorname{out}_n(X,A)\right\|_2\right] \\
&\quad + O\left(\tau_v^2 \|A\|_1 \mathcal{B}_\mathcal{F} + \tau_v \widetilde{\alpha}^2 \|A\|_1^2\right) \\
&\quad + O\left(\tau_v \|A\|_1 + \alpha\mathcal{L}_\mathcal{G}\|A\|_1\right) \cdot \left(\tau_v \|A\|_1 \mathcal{B}_\mathcal{F} + \alpha\mathcal{B}_{\mathcal{F}\circ\mathcal{G}}\right)
\end{aligned}
\tag{63}
$$

Using $\mathcal{B}_{\mathcal{F}\circ\mathcal{G}} \le \sqrt{k} p_\mathcal{G} \mathcal{C}_s(\mathcal{G}, \|A\|_1 \mathcal{B}_\mathcal{F})(\|A\|_1 \mathcal{B}_\mathcal{F})^2 \mathcal{B}_\mathcal{F} \le \frac{\tau_v}{\alpha}\mathcal{B}_\mathcal{F}$, so we have

$$
\begin{aligned}
\mathop{\mathbb{E}}_{n\in\mathcal{V},(X,y_n)\sim\mathcal{D}} \|Err\|_2 &\le \mathop{\mathbb{E}}_{n\in\mathcal{V},(X,y_n)\sim\mathcal{D}} \left[O\left(\tau_v \|A\|_1 + \alpha\mathcal{L}_\mathcal{G}\|A\|_1\right) \cdot \left\|\mathcal{H}(x) - \operatorname{out}_n(X,A)\right\|_2\right] \\
&\quad + O\left(\tau_v^2 \|A\|_1^2 \mathcal{B}_\mathcal{F} + \tau_v \widetilde{\alpha}^2 \|A\|_1^2 + \alpha\tau_v \|A\|_1^2 \mathcal{L}_\mathcal{G}\mathcal{B}_\mathcal{F}\right)
\end{aligned}
\tag{64}
$$

**Claim C.6.** *In the setting of Lemma C.1, if $\tau_v \|A\|_1 \le \frac{1}{polylog(m)}$, we have*

$$
\left\|\operatorname{out}_n^1(X,A) - \mathcal{F}_n(X)\right\|_2 \le 2\left\|\operatorname{out}_n^1(X,A) - \mathcal{H}_n(X,A)\right\|_2 + \alpha\mathcal{B}_{\mathcal{F}\circ\mathcal{G}} + \widetilde{O}\left(\tau_v \|A\|_1 \mathcal{B}_\mathcal{F}\right)
\tag{65}
$$

Proof: Using 34 and $\left\|\mathcal{G}(\mathcal{F}(X), a_n)\right\|_2 \le \mathcal{B}_{\mathcal{F}\circ\mathcal{G}}$, we have

$$
\begin{aligned}
\left\|\operatorname{out}_n^1(X,A) - \mathcal{F}_n(X)\right\|_2 &\le \left\|\operatorname{out}_n^1(X,A) - \mathcal{H}_n(X,A)\right\|_2 + \alpha\mathcal{B}_{\mathcal{F}\circ\mathcal{G}} \\
&\quad + \widetilde{O}\left(\tau_v \|A\|_1 \left(\left\|\operatorname{out}_n^1(X, a_n) - \mathcal{F}_n(X,A)\right\|_2 + \mathcal{B}_\mathcal{F}\right)\right)
\end{aligned}
\tag{66}
$$

Using $\tau_v \|A\|_1$ small enough, we have

$$
\left\|\operatorname{out}_n^1(X,A) - \mathcal{F}_n(X)\right\|_2 \le 2\left\|\operatorname{out}_n^1(X,A) - \mathcal{H}_n(X,A)\right\|_2 + \alpha\mathcal{B}_{\mathcal{F}\circ\mathcal{G}} + \widetilde{O}\left(\tau_v \|A\|_1 \mathcal{B}_\mathcal{F}\right)
\tag{67}
$$

### C.4 Proof of Theorem 3.2

Proof: Using (50) and Claim C.5, in iteration $t$, we have

$$
\begin{aligned}
&\langle \nabla_{\mathbf{W},\mathbf{V}} \operatorname{Obj}_n(\mathbf{W}_t,\mathbf{V}_t)),(\mathbf{W}_t - \mathbf{W}^*,\mathbf{V}_t - \mathbf{V}^*))\rangle \\
&= \langle y_n - \operatorname{out}(\mathbf{W}_t,\mathbf{V}_t), f(\mathbf{W}^* - \mathbf{W}) + g(\mathbf{V}^* - \mathbf{V})\rangle \\
&= \langle y_n - \operatorname{out}_n(\mathbf{W}_t,\mathbf{V}_t), \mathcal{H}_{n,A^*}(X) - \operatorname{out}_n(\mathbf{W}_t,\mathbf{V}_t) + Err_t\rangle
\end{aligned}
\tag{68}
$$

We also have

$$
\begin{aligned}
\|W_{t+1} - W^*\|_F^2 &= \|W_t - \eta_w \nabla_W \operatorname{Obj}_n(W_t) - W^*\|_F^2 \\
&= \|W_t - W^*\|_F^2 - 2\eta_w \langle \nabla_W \operatorname{Obj}_n(W_t), W_t - W^*\rangle \\
&\quad + \eta_w^2 \|\nabla_W \operatorname{Obj}_n(W_t)\|_F^2,
\end{aligned}
\tag{69}
$$

$$
\begin{aligned}
\|V_{t+1} - V^*\|_F^2 &= \|V_t - \eta_v \nabla_V \operatorname{Obj}_n(V_t) - V^*\|_F^2 \\
&= \|V_t - V^*\|_F^2 - 2\eta_v \langle \nabla_V \operatorname{Obj}_n(V_t), V_t - V^*\rangle \\
&\quad + \eta_v^2 \|\nabla_V \operatorname{Obj}_n(V_t)\|_F^2
\end{aligned}
\tag{70}
$$

Using Algorithm 1, we have $\mathbf{W}_{t+1} = \mathbf{W}_t - \eta_w \nabla_{\mathbf{W}} \operatorname{Obj}_n(\mathbf{W}_t,\mathbf{V}_t)$ and $\mathbf{V}_{t+1} = \mathbf{V}_t - \eta_v \nabla_{\mathbf{V}} \operatorname{Obj}_n(\mathbf{W}_t,\mathbf{V}_t)$, so we have

$$
\begin{aligned}
&\langle \nabla_{\mathbf{W},\mathbf{V}} \operatorname{Obj}_t(\mathbf{W}_t,\mathbf{V}_t),(\mathbf{W} - \mathbf{W}^*,\mathbf{V} - \mathbf{V}^*))\rangle \\
&= \underbrace{\frac{\eta_w}{2}\|\nabla_{\mathbf{W}} \operatorname{Obj}_t(\mathbf{W}_t,\mathbf{V}_t)\|_F^2 + \frac{\eta_v}{2}\|\nabla_{\mathbf{V}} \operatorname{Obj}_t(\mathbf{W}_t,\mathbf{V}_t)\|_F^2}_{\heartsuit} \\
&\quad + \frac{1}{2\eta_w}\|\mathbf{W}_t - \mathbf{W}^*\|_F^2 - \frac{1}{2\eta_w}\|\mathbf{W}_{t+1} - \mathbf{W}^*\|_F^2 + \frac{1}{2\eta_v}\|\mathbf{V}_t - \mathbf{V}^*\|_F^2 - \frac{1}{2\eta_v}\|\mathbf{V}_{t+1} - \mathbf{V}^*\|_F^2
\end{aligned}
\tag{71}
$$

Using Claim C.4 and change all $A$ to $A^*$, we have

$$
\begin{aligned}
\heartsuit &\leq O\left(\eta_w \sigma_v^2 + \eta_v \tau_w^2\right) \cdot \|A\|_1^2 \|y_n - \operatorname{out}_n(X,A^*)\|_2^2 \\
&\leq O\left(\eta_w \sigma_v^2 + \eta_v \tau_w^2\right) \cdot \|A\|_1^2 \left(\|\mathcal{H}_{n,A^*}(X) - \operatorname{out}_n(X,A^*)\|_2^2 + \|\mathcal{H}_{n,A^*}(X) - y_n\|_2^2\right)
\end{aligned}
\tag{72}
$$

Therefore, as long as $O\left(\eta_w \sigma_v^2 + \eta_v \tau_w^2\right) \leq 0.1$, it satisfies

$$
\begin{aligned}
\frac{1}{4}\|\mathcal{H}_{n,A^*}(X) - \operatorname{out}_n(X,A^*)\|_2^2 \leq{}& 2\|\operatorname{Err}_t\|_2^2 + 4\|\mathcal{H}_{n,A^*}(X) - y_n\|_2^2 \\
&+ \frac{1}{2\eta_w}\|\mathbf{W}_t - \mathbf{W}^*\|_F^2 - \frac{1}{2\eta_w}\|\mathbf{W}_{t+1} - \mathbf{W}^*\|_F^2 \\
&+ \frac{1}{2\eta_v}\|\mathbf{V}_t - \mathbf{V}^*\|_F^2 - \frac{1}{2\eta_v}\|\mathbf{V}_{t+1} - \mathbf{V}^*\|_F^2
\end{aligned}
\tag{73}
$$

After telescoping for $t = 0,1,\ldots,T_0 - 1$,

$$
\begin{aligned}
&\frac{\|\mathbf{W}_{T_0} - \mathbf{W}^*\|_F^2}{2\eta_w T_0} + \frac{\|\mathbf{W}_{T_0} - \mathbf{V}^*\|_F^2}{2\eta_v T_0} + \frac{1}{2T_0}\sum_{t=0}^{T_0-1}\|\mathcal{H}_{n,A^*}(X) - \operatorname{out}_n(X,A^*)\|_2^2 \\
&\leq \frac{\|\mathbf{W}^*\|_F^2}{2\eta_w T_0} + \frac{\|\mathbf{V}^*\|_F^2}{2\eta_v T_0} + \frac{O(1)}{T_0}\sum_{t=0}^{T_0-1}\|Err_t\|_2^2 + \|\mathcal{H}_{n,A^*}(X) - y_t\|_2^2.
\end{aligned}
\tag{74}
$$

Using $O\left(\tau_v \|A\|_1 + \alpha \mathcal{L}_{\mathcal{G}}\right) \leq 0.1$, we have

$$
\frac{1}{4T}\sum_{t=0}^{T-1} \underset{n \in \mathcal{V},(X,y_n)\sim\mathcal{Z}}{\mathbb{E}} \|\mathcal{H}_{n,A^*}(X) - \operatorname{out}_n(X,A^*)\|_2^2 \leq \frac{\|\mathbf{W}^*\|_F^2}{2\eta_w T} + \frac{\|\mathbf{V}^*\|_F^2}{2\eta_v T} + O(\operatorname{OPT} + \epsilon_0)
\tag{75}
$$

where

$$
\begin{aligned}
\epsilon_0 &= \Theta \left( \widetilde{\alpha}^2 \tau_v \left\| A^* \right\|_1^2 + \tau_v^2 \left\| A^* \right\|_1 \mathcal{B}_{\mathcal{F}} + \alpha \tau_v \left\| A^* \right\|_1 \mathcal{L}_{\mathcal{G}} \mathcal{B}_{\mathcal{F}} \right)^2 \\
&= \widetilde{\Theta} \left( \widetilde{\alpha}^2 \tau_v \left\| A^* \right\|_1^2 + \alpha^2 (k p_{\mathcal{G}} \mathcal{C}_s (\mathcal{G}, \mathcal{B}_{\mathcal{F}} \left\| A^* \right\|_1)^2 (\mathcal{B}_{\mathcal{F}} \left\| A^* \right\|_1)^3 \right. \\
&\qquad \left. + \alpha^2 k p_{\mathcal{G}} \mathcal{C}_s (\mathcal{G}, \mathcal{B}_{\mathcal{F}} \left\| A^* \right\|_1)(\mathcal{B}_{\mathcal{F}} \left\| A^* \right\|_1)^3 \left\| A^* \right\|_1 \right)^2 \\
&= \widetilde{\Theta} \left( \alpha^4 \left( p_{\mathcal{G}} \mathcal{C}_s (\mathcal{G}, \mathcal{B}_{\mathcal{F}} \left\| A^* \right\|_1) \right)^4 \left( \left\| A^* \right\|_1 \mathcal{B}_{\mathcal{F}} \right)^6 \right)
\end{aligned}
\tag{76}
$$

In practice, for computational efficiency, we use the sampled adjacency matrix $A^t$ in the learning network, so we should consider the discrepancy between the target function and the practical output

$$
\begin{aligned}
\left\| \mathcal{H}_{n,A^*}(X) - \mathrm{out}_n \left( X, A^{1t}, A^{2t} \right) \right\|_2^2 \leq {} & \left\| \mathcal{H}_{n,A^*}(X) - \mathrm{out}_n \left( X, A^* \right) \right\|_2^2 \\
& + \left\| \mathrm{out}_n \left( X, A^{1t}, A^{2t} \right) - \mathrm{out}_n \left( X, A^* \right) \right\|_2^2
\end{aligned}
\tag{77}
$$

We have already considered $\left\| \mathcal{H}_{n,A^*}(X) - \mathrm{out}_n \left( X, A^* \right) \right\|_2^2$ and

$$
\begin{aligned}
\left\| \mathrm{out}_n \left( X, A^{1t}, A^{2t} \right) - \mathrm{out}_n \left( X, A^* \right) \right\|_2 \leq {} & \left\| C\sigma(WXa_n^{1t}) - C\sigma W X a_n^* \right\|_2 \\
& + \left\| C\sigma(V \, \mathrm{out}_n^1(XA^{1t})a_n^{2t}) - C\sigma(V \, \mathrm{out}_n^1(XA^*)a_n^*) \right\|_2
\end{aligned}
\tag{78}
$$

Using (30), we have

$$
\left\| C\sigma(WXa_n^{1t}) - C\sigma W X a_n^* \right\|_2 \leq \tau_w \left\| X a_n^{1t} - X a_n^* \right\|_2 \leq \left\| \mathrm{Err}_t \right\|_2
\tag{79}
$$

For the above equation to hold, it requires

$$
\left\| A^{1t} - A^* \right\|_1 \leq \left\| \frac{Err_t}{\tau_w} \right\|_2
\tag{80}
$$

Using $\left\| A^* \right\|_1 \leq O(1)$ and (34), we have

$$
\begin{aligned}
\left\| C\sigma(V \, \mathrm{out}_n^1(XA^{1t})a_n^{2t}) - C\sigma(V \, \mathrm{out}_n^1(XA^*)a_n^*) \right\|_2 &\leq \tau_v \left\| \mathrm{out}_n^1(XA^{1t})a_n^{2t} - \mathrm{out}_n^1(XA^*)a_n^* \right\|_2 \\
&\leq \tau_v \tau_w \left\| A^{2t} - A^* \right\|_1 \leq \left\| \mathrm{Err}_t \right\|_2
\end{aligned}
\tag{81}
$$

For the above equation to hold, it requires $\left\| A^{2t} - A^* \right\|_1 \leq \left\| \frac{Err_t}{\tau_v \tau_w} \right\|_2$.

Under assumptions of Lemma C.7, with high probability, we can ensure $\left\| A^{1t} - A^* \right\|_2 \leq \left\| \frac{Err_t}{\tau_w} \right\|_2$, $\left\| A^{2t} - A^* \right\|_1 \leq \left\| \frac{Err_t}{\tau_v \tau_w} \right\|_2$.

Using $\left\| \mathbf{W}^* \right\|_F \leq \tau_w/10, \left\| \mathbf{V}^* \right\|_F \leq \tau_v/10$ and $\epsilon \geq \mathrm{OPT} + \epsilon_0$, we have

$$
\frac{1}{T} \sum_{t=0}^{T-1} \mathop{\mathbb{E}}_{n \in \mathcal{V}, (X, y_n) \sim \mathcal{D}} \left\| \mathcal{H}_{n,A^*}(X) - \mathrm{out}_n \left( X, A^{1t}, A^{2t} \right) \right\|_2^2 \leq O(\epsilon)
\tag{82}
$$

as long as $T \geq \Omega \left( \frac{\tau_w^2/\eta_w + \tau_v^2/\eta_v}{\epsilon} \right)$.

Finally, we should check $\left\| \mathbf{W}_t \right\|_F \leq \tau_w$ and $\left\| \mathbf{V}_t \right\|_F \leq \tau_v$ hold.

$$
\frac{\left\| \mathbf{W}_{T_0} - \mathbf{W}^* \right\|_F^2}{2\eta_w T_0} + \frac{\left\| \mathbf{V}_{T_0} - \mathbf{V}^* \right\|_F^2}{2\eta_v T_0} \leq \frac{\left\| \mathbf{W}^* \right\|_F^2}{2\eta_w T_0} + \frac{\left\| \mathbf{V}^* \right\|_F^2}{2\eta_v T_0} + O(\epsilon) + \widetilde{O} \left( \frac{\tau_w \left\| A^* \right\|_1}{\sqrt{T_0}} \right)
\tag{83}
$$

Using the relationship $\frac{\tau_w^2}{\eta_w} = \frac{\tau_v^2}{\eta_v}$, we have

$$
\frac{\left\| \mathbf{W}_{T_0} \right\|_F^2}{\tau_w^2} + \frac{\left\| \mathbf{V}_{T_0} \right\|_F^2}{\tau_v^2} \leq \frac{4 \left\| \mathbf{W}^* \right\|_F^2}{\tau_w^2} + \frac{4 \left\| \mathbf{V}^* \right\|_F^2}{\tau_v^2} + 0.1 + \widetilde{O} \left( \frac{\eta_w \left\| A^* \right\|_1 \sqrt{T_0}}{\tau_w} \right)
\tag{84}
$$

Therefore, choosing $T = \widetilde{\Theta}\left(\frac{\tau_w^2}{\|A^*\|_1 \min\{1,\epsilon^2\}}\right)$ and $\eta_w = \widetilde{\Theta}(\min\{1,\epsilon\}) \leq 0.1$, we can ensure $\frac{\|\mathbf{W}_{T_0}\|_F^2}{\tau_w^2} + \frac{\|\mathbf{V}_{T_0}\|_F^2}{\tau_v^2} \leq 1$.

## C.5 Graph sampling

**Lemma C.7.** *Given a graph $G$ with the minimum degree $\delta(G) \geq \Omega(\left\|\frac{\tau_w}{Err_t}\right\|_2)$, in iteration $t$, for the first layer $A^{1t}$ is generated from the sampling strategy with the sampling probability $p_{ij}^1 \leq O(\frac{\sqrt{d_i d_j}\|Err_t\|_2}{n_{ij}\tau_w})$ and for the second layer $A^{2t}$ is generated from the sampling strategy with the sampling probability $p_{ij}^2 \leq O(\frac{\sqrt{d_i d_j}\|Err_t\|_2}{n_{ij}\tau_w\tau_v})$, we have*

$$\Pr\left[\left\|A^{1t} - A^*\right\|_1 \leq O(\left\|\frac{Err_t}{\tau_w}\right\|_2)\right] \leq e^{-\Omega(\|Err_t\|_2\sqrt{d_i d_j}/\tau_w)} \tag{85}$$

$$\Pr\left[\left\|A^{2t} - A^*\right\|_1 \leq O(\left\|\frac{Err_t}{\tau_w\tau_v}\right\|_2)\right] \leq e^{-\Omega(\|Err_t\|_2\sqrt{d_i d_j}/\tau_w\tau_v)} \tag{86}$$

Proof: The difference between $A_{B_{ij}}^t$ and $A_{B_{ij}}^*$ is

$$\Delta_{B_{ij}} = \left\|A_{B_{ij}}^t - A_{B_{ij}}^*\right\| = \sum_{i=1}^{n_{ij}}(a_{ij}^t - a_{ij}^*) \tag{87}$$

where $n_{ij}$ is the number of elements in $A_{B_{ij}}^*$ and $(a_{ij}^t - a_{ij}^*)$ are iid, with $\mu_{ij} = \mathbb{E}[\Delta_{B_{ij}}] = n_{ij}p_{ij}\frac{1}{\sqrt{d_i d_j}}$. The Moment-generating function of $(a_{ij}^t - a_{ij}^*)$ is

$$\begin{aligned}
M_{(a_{ij}^t - a_{ij}^*)}(s) &= \mathbb{E}\left[e^{s(a_{ij}^t - a_{ij}^*)}\right] \\
&= e^{s\frac{1}{\sqrt{d_i d_j}}}p_{ij} + e^{s\cdot 0}(1 - p_{ij}) \\
&= 1 + p_{ij}\left(e^{\frac{s}{\sqrt{d_i d_j}}} - 1\right) \\
&\leq exp\left(e^{\frac{s}{\sqrt{d_i d_j}}} - 1\right)
\end{aligned} \tag{88}$$

Thus, for any $t > 0$, using Markov's inequality and the definition of MGF, we have

$$\begin{aligned}
\mathbb{P}(\Delta_{B_{ij}} \geq k) &\leq \min_{s>0} \frac{\prod_{i=1}^{n_{ij}} M_{(a_{ij}^t - a_{ij}^*)}(s)}{e^{tk}} \\
&= \min_{t>0} \frac{e^{\mu\sqrt{d_i d_j}\left(e^{\frac{s}{\sqrt{d_i d_j}}} - 1\right)}}{e^{tk}}
\end{aligned} \tag{89}$$

If $0 \leq \delta_{ij} \leq 1$, we plug in $k_{ij} = (1 + \delta_{ij})\mu_{ij}$ and the optimal value of $s_{ij} = \sqrt{d_i d_j}\ln(1 + \epsilon_{ij})$ to the above equation:

$$\mathbb{P}(\Delta_{B_{ij}} \geq (1 + \delta_{ij})\mu_{ij}) \leq \left(\frac{e^{\epsilon_{ij}}}{(1 + \epsilon_{ij})^{(1+\epsilon_{ij})}}\right)^{\mu_{ij}\sqrt{d_i d_j}} \leq \exp\left(\frac{-\epsilon_{ij}^2\mu_{ij}\sqrt{d_i d_j}}{3}\right) \tag{90}$$

$$\begin{aligned}
(1 + \delta_{ij})^{(1+\delta_{ij})} &= exp[(1 + \delta_{ij})ln(1 + \delta_{ij})] \\
&= exp(\delta_{ij} + \frac{\delta_{ij}^2}{2} - \frac{\delta_{ij}^3}{6} + o(\delta_{ij}^4)) \geq exp(\delta_{ij} + \frac{\delta_{ij}^2}{2} - \frac{\delta_{ij}^3}{6})
\end{aligned} \tag{91}$$

Let $\delta_{ij} = 1$, $\mu_{ij} = \Theta(Err_t)$, and $d_i \geq \Omega(\frac{1}{Err_t})$, we have $p_{ij} \leq O(\frac{\sqrt{d_i d_j} Err_t}{n_{ij}})$.

## C.6 Sample Complexity

**Lemma C.8.** *Given a graph $G$ with $|V(G)| = N$, if the maximum degree $\Delta(G) \leq O((N\epsilon^2)^{\frac{1}{4}})$ and sample complexity $\Omega \geq O(\frac{\Delta(G)^2(\tau_w\|A\|_1)^2 \log N}{\epsilon^2})$, with probability $1 - N^{-\tau_w\|A\|_1}$, we have*
$$\left| \mathop{\mathbb{E}}_{n \in \mathcal{V}, (X, y_n) \sim \mathcal{Z}} \left\| y_n - \text{out}_n \left( X, A^{1t}, A^{2t} \right) \right\|_2 - \mathop{\mathbb{E}}_{n \in \mathcal{V}, (X, y_n) \sim \mathcal{D}} \left\| y_n - \text{out}_n \left( X, A^{1t}, A^{2t} \right) \right\|_2 \right| \leq \epsilon.$$

Proof: For the set of samples Z define

$$\mathop{\mathbb{E}}_{n \in \mathcal{V}, (X, y_n) \sim \mathcal{Z}} \left\| y_n - \text{out}_n \left( X, A^{1t}, A^{2t} \right) \right\|_2 = \frac{1}{\Omega} \sum_{n=1}^{\Omega} \left\| y_n - \text{out}_n \left( X, A^{1t}, A^{2t} \right) \right\|_2 \tag{92}$$

Denote the generalization error as

$$\left| \mathop{\mathbb{E}}_{n \in \mathcal{V}, (X, y_n) \sim \mathcal{Z}} \left\| y_n - \text{out}_n \left( X, A^{1t}, A^{2t} \right) \right\|_2 - \mathop{\mathbb{E}}_{n \in \mathcal{V}, (X, y_n) \sim \mathcal{D}} \left\| y_n - \text{out}_n \left( X, A^{1t}, A^{2t} \right) \right\|_2 \right|$$

$$= \left| \mathop{\mathbb{E}}_{n \in \mathcal{V}, (X, y_n) \sim \mathcal{Z}} \left\| \text{out}_n \left( X, A^{1t}, A^{2t} \right) \right\|_2 - \mathop{\mathbb{E}}_{n \in \mathcal{V}, (X, y_n) \sim \mathcal{D}} \left\| \text{out}_n \left( X, A^{1t}, A^{2t} \right) \right\|_2 \right|$$

By Hoeffding's inequality and $\|\text{out}_n(X, a_n)\|_2 \leq O(\tau_w \|A\|_1)$ , we have

$$\mathbb{E} \left[ e^{s \left| \mathop{\mathbb{E}}_{n \in \mathcal{V}, (X, y_n) \sim \mathcal{Z}} \left\| \text{out}_n \left( X, A^{1t}, A^{2t} \right) \right\|_2 - \mathop{\mathbb{E}}_{n \in \mathcal{V}, (X, y_n) \sim \mathcal{D}} \left\| \text{out}_n \left( X, A^{1t}, A^{2t} \right) \right\|_2 \right|} \right] \leq e^{\frac{(s\tau_w \|A\|_1)^2}{8}} \tag{93}$$

Define maximum degree of G is $\Delta(G)$. It is easy to know that $\left\| \text{out}_n \left( X, A^{1t}, A^{2t} \right) \right\|_2$ is dependent with at most its second order neighbor, so the maximum number of nodes related with $\left\| \text{out}_n \left( X, A^{1t}, A^{2t} \right) \right\|_2$ is $\Delta(G)^2$. By Lemma 7 in Shuai, we have

$$\mathbb{E} e^{s \sum_{n=1}^{\Omega} \left\| \text{out}_n \left( X, A^{1t}, A^{2t} \right) \right\|_2} \leq e^{\Delta(G)^2 (s\tau_w \|A\|_1)^2 \Omega/8} \tag{94}$$

$$\mathbb{P} \left( \left| \mathop{\mathbb{E}}_{n \in \mathcal{V}, (X, y_n) \sim \mathcal{Z}} \left\| \text{out}_n \left( X, A^{1t}, A^{2t} \right) \right\|_2 - \mathop{\mathbb{E}}_{n \in \mathcal{V}, (X, y_n) \sim \mathcal{D}} \left\| \text{out}_n \left( X, A^{1t}, A^{2t} \right) \right\|_2 \right| \geq \epsilon \right)$$
$$\leq \exp \left( \Delta(G)^2 (s\tau_w \|A\|_1)^2 \Omega/8 - s\epsilon\Omega \right) \tag{95}$$

Let $s = \frac{4\epsilon}{\Delta(G)^2 (\tau_w \|A\|_1)^2}$ and $\epsilon = (\tau_w \|A\|_1)^2 \sqrt{\frac{\Delta(G)^4 \log N}{\Omega}}$

$$\mathbb{P} \left( \left| \mathop{\mathbb{E}}_{n \in \mathcal{V}, (X, y_n) \sim \mathcal{Z}} \left\| \text{out}_n \left( X, A^{1t}, A^{2t} \right) \right\|_2 - \mathop{\mathbb{E}}_{n \in \mathcal{V}, (X, y_n) \sim \mathcal{D}} \left\| \text{out}_n \left( X, A^{1t}, A^{2t} \right) \right\|_2 \right| \geq \epsilon \right)$$
$$\leq \exp \left( -(\tau_w\|A\|_1)^2 \log N \right)$$
$$\leq N^{-\tau_w\|A\|_1} \tag{96}$$

with

$$\Omega \geq O(\frac{\Delta(G)^2 (\tau_w \|A\|_1)^2 \log N}{\epsilon^2}) \tag{97}$$

