matrix $A_{B_{ij}} \in \mathbb{R}^{N_i \times N_j}$ denote a submatrix of $A$ with rows in group $i$ and columns corresponding to group $j$. Then all entries in $A_{B_{ij}}$ are in the order of $\frac{1}{\sqrt{d_i d_j}}$. Note that a relatively smaller entry in $A$ corresponds to an edge between relatively higher-degree nodes. Let $p_{ij}^k$ in $[0, 1]$ ($k = 1, 2$) reflect the probability of remaining weights on smaller entries in $A_{B_{ij}}$ (i.e., the probability of pruning weights on smaller entries in $A_{B_{ij}}$ is $1 - p_{ij}^k$) in the first and second hidden layers, respectively. Our sparsification strategy can be described as follows: at each iteration, for each submatrix $A_{B_{ij}}$,

(1) if $i > j$, each of the top[2] $d_1 \sqrt{\frac{d_i}{d_j}}$ large-weight edges $A_{ij}$ in $A_{B_{ij}}$ is retained independently with probability $1 - p_{ij}^k$. The remaining entries in $A_{B_{ij}}$ are retained independently with a probability of $p_{ij}^k$.

(2) if $i \leq j$, each of the top $d_1$ largest $A_{ij}$ in $A_{B_{ij}}$ are retained with probability $1 - p_{ij}^k$. The remaining entries in $A_{B_{ij}}$ are retained independently with a probability of $p_{ij}^k$.

We allow the pruning rates to vary in different layers and will quantify how these weights affect generalization differently. When others are fixed, a small $p_{ij}^k$ indicates retaining primarily large-weight edges, while a large $p_{ij}^k$ indicates retaining less large-weight edges.

To see why this sparsification strategy prioritizes low-degree edges, consider the pruned edges between a fixed group $i$ and other groups $j$. Assume $p_{ij}^k$ are the same for all groups for simplicity. If $d_j < d_{j'}$ for two groups $j$ and $j'$, then $d_1 \sqrt{\frac{d_i}{d_j}} > d_1 \sqrt{\frac{d_i}{d_{j'}}}$, indicating that more lower-degree edges (large-weight edges) connecting groups $i$ and $j$ are retained, compared with edges connecting groups $i$ and $j'$.

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

\end{aligned}
\tag{12}
$$

Note that $\widetilde{\Theta}\left(\alpha \mathcal{C}_s(\mathcal{G})\right) < 1$ (see Table 3 in Appendix for the parameters) , then the upper bound for $p_{ij}^2$ is higher than that for $p_{ij}^1$ in the assumption. That means the pruning for the first hidden layer must focus more on low-degree edges (large-weight edges), while such a requirement is relaxed in the second layer. Then, (12) indicates that a larger deviation of the sparsified matrix $A^t$ from $A^*$ can be tolerated in the second hidden layer compared with the first layer. Lemma 3.1 reveals that the jumping connection allows for a more flexible sparsification strategy in deeper layers.

We will show the learned model can achieve an error close to $O(\text{OPT})$. Our main theorem can be sketched as follows,

**Theorem 3.2.** *For $\epsilon_0 = \widetilde{\Theta}(\alpha^4 \mathcal{C}_s(\mathcal{G})^4) < 1$ and $\epsilon = 10 \cdot \text{OPT} + \epsilon_0$, suppose pruning probability $p_{ij}^1 \leq \widetilde{\Theta}(\frac{\sqrt{d_i d_j} \epsilon_0}{N_i N_j \mathcal{C}_s(\mathcal{F})})$ and $p_{ij}^2 \leq \widetilde{\Theta}(\frac{\sqrt{d_i d_j} \epsilon_0}{N_i N_j \alpha \mathcal{C}_s(\mathcal{F}) \mathcal{C}_s(\mathcal{G})})$ , there exist $M_0 = \text{poly}\left(C_s(\mathcal{F}), C_s(\mathcal{G}), \alpha^{-1}\right)$, $T_0 = \widetilde{\Theta}\left(\frac{\mathcal{C}_s(\mathcal{F})^2}{\|A^*\|_1 \min\{0.1, \epsilon^2\}}\right)$ and $N_0 = \widetilde{\Theta}(\Delta^4 \mathcal{C}_s(\mathcal{F})^2 \|A^*\|_1^4 \log N \epsilon^{-2})$ such that for every $m \geq M_0$, $T \geq T_0$ and $|\Omega| \geq N_0$, with high probability, the SGD algorithm satisfies*

$$
\frac{1}{T} \sum_{t=0}^{T-1} \mathop{\mathbb{E}}_{\substack{(X, y_n) \sim \mathcal{D} \\ n \in \mathcal{V}}} \left\| y_n - \text{out}_n\left(X, A^{1t}, A^{2t}; \mathbf{W}_t, \mathbf{V}_t\right) \right\|_2^2 \leq \epsilon
\tag{13}
$$

As a sanity check, when the concept class enlarges, OPT decreases. Theorem 3.2 shows that the required number of neurons $M_0$ (model complexity) and labels $N_0$ (sample complexity) both increase accordingly. $p_{ij}^1$ and $p_{ij}^2$ decreasing means that we should retain more large-weight edges. Thus, our theoretical bounds match the intuition that a larger model, more labels, and more high-weight edges improve the prediction accuracy. $N_0$ is in the order of $\log N$, indicating that the unknown labels can be accurately predicted from partial labels. Moreover, when $\|A^*\|_1$ increases, $C_\varepsilon$ and $\mathcal{C}_s$, and $\epsilon_0$ all increase. Theorem 3.2 indicates the model complexity $M_0$, $N_0$, and the generalization error $\epsilon$ all increasing, indicating worse prediction performance.

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

$$+ m^{3/4} \cdot (\mathfrak{C}_\varepsilon(\mathcal{F}, \|A\|_1) \|A\|_1)^2, \tag{37}$$
$$\leq 2m(\mathfrak{C}_\mathfrak{s}(\mathcal{F}, \|A\|_1) \|A\|_1)^2.$$

Putting this back to (36) we have $\|\mathbf{W}^*\|_F^2 \leq \widetilde{O}\left( \frac{k^2 p^2 (\mathfrak{C}_\mathfrak{s}(\mathcal{F}, \|A\|_1) \|A\|_1)^2}{m} \right)$.

**Lemma C.3.** *Under the assumptions of Lemma C.1, suppose $\alpha \in (0, 1)$ and $\widetilde{\alpha} = \frac{\alpha}{k(p_\mathcal{F} \mathfrak{C}_\mathfrak{s}(\mathcal{F}, \|A\|_1) + p_\mathcal{G} \mathfrak{C}_\mathfrak{s}(\mathcal{G}, \|A\|_1))}$, there exist $M = \text{poly}\left( \mathfrak{C}_{\widetilde{\alpha}}(\mathcal{F}, \|A\|_1), \mathfrak{C}_{\widetilde{\alpha}}(\mathcal{G}, \|A\|_1), \|A\|_1, \widetilde{\alpha}^{-1} \right)$ satisfying that for every $m \geq M, \|\mathbf{W}^*\|_F \leq \widetilde{O}(kp_\mathcal{F} \mathfrak{C}_\mathfrak{s}(\mathcal{F}))$ and $\|\mathbf{V}^*\|_F \leq \widetilde{O}(\widetilde{\alpha} kp_\mathcal{G} \mathfrak{C}_\mathfrak{s}(\mathcal{G}))$ with high probability*

1.
$$\mathbb{E}_{n \in \mathcal{V}, (X, y_n) \sim \mathcal{D}} \left[ \left\| \mathbf{CD}_{\mathbf{W_0}}^n \mathbf{W}^* X a_n - \mathcal{F}_n(X, A) \right\|_2 \right] \leq \widetilde{\alpha}^2 \|A\|_1 \tag{38}$$

2.
$$\left\| \mathbf{CD}_{\mathbf{V_0}}^n \mathbf{V}^* \text{out}^1(X, A) a_n - \alpha \mathcal{G}_n \left( \text{out}^1(X, A), A \right) \right\|_2 \leq \widetilde{\alpha}^2 \cdot \left\| \text{out}_n^1(X, A) \right\|_2 \|A\|_1 \tag{39}$$

3.
$$\mathbb{E}_{n \in \mathcal{V}, (X, y_n) \sim \mathcal{D}} \left[ \left\| \mathbf{CD}_{\mathbf{W}}^n \mathbf{W}^* X a_n - \mathcal{F}_n(X, A) \right\|_2 \right] \leq O(\widetilde{\alpha}^2 \|A\|_1) \tag{40}$$

4.
$$\left\| \mathbf{CD}_{\mathbf{V}}^n \mathbf{V}^* \text{out}^1(X, A) a_n - \alpha \mathcal{G}_n \left( \text{out}^1(X, A), A \right) \right\|_2$$
$$\leq \left( \widetilde{\alpha}^2 + O\left( \tau_v \left( \frac{\tau_v}{\sigma_v} \right)^{1/3} \right) \right) \left\| \text{out}_n^1(X, A) \right\|_2 \|A\|_1 \tag{41}$$

5.
$$\mathbb{E}_{n\in\mathcal{V},(X,y_n)\sim\mathcal{D}}\left[\left\|\mathbf{C}D^n_{\mathbf{W_0}+\mathbf{w}}(\mathbf{W}^*-\mathbf{W})Xa_n-(\mathcal{F}_n(X,A)-\mathrm{out}^1_n(X,A))\right\|_2\right]\leq\widetilde{\alpha}^2\left\|A\right\|_1 \tag{42}$$

Proof:

1. Using Lemma C.2, we can find a $\mathbf{W}^*$ satisfying $\left\|\mathbf{C}D^n_{\mathbf{W_0}+\mathbf{w}}\mathbf{W}^*Xa_n-\mathcal{F}_n(X,A)\right\|_2$ small enough with probability at least $1-e^{-\Omega(\sqrt{m})}$.

2. Using Lemma C.2 and $\left\|\mathrm{out}^1(X,A)a_n\right\|_2\leq\|\mathrm{out}^1_n(X,A)\|_2\left\|A\right\|_1$, we can easily prove it.

3. $\left\|\mathbf{W}^*Xa_n\right\|_2 \leq O\left(\left\|\mathbf{W}^*\right\|_F\left\|Xa_n\right\|_2\right) \leq O\left(\tau_w\left\|A\right\|_1\right).$ $\left\|\mathbf{C}(D^n_{\mathbf{W}}-D^n_{\mathbf{W_0}})\mathbf{W}^*Xa_n\right\|_2 \leq O\left(\sqrt{s}\tau_w\left\|A\right\|_1/\sqrt{m}\right)$ where $s$ is the maximum sparsity of $(D^n_{\mathbf{W}}-D^n_{\mathbf{W_0}})$. Using (25), we know $s\leq O\left((\frac{\tau_w}{\sigma_w})^{2/3}m^{2/3}\right)$. This, combining with (38) gives

$$\mathbb{E}_{n\in\mathcal{V},(X,y_n)\sim\mathcal{D}}\left[\left\|\mathbf{C}D^n_{\mathbf{W}}\mathbf{W}^*Xa_n-\mathcal{F}_n(X,A)\right\|_2\right]\leq\widetilde{\alpha}^2\left\|A\right\|_1+O\left(\tau_w(\frac{\tau_w}{\sigma_w})^{1/3}/m^{1/6}\right)$$
$$\leq O(\widetilde{\alpha}^2\left\|A\right\|_1) \tag{43}$$

4. Using (28) and $\left\|\mathbf{V}^*\mathrm{out}^1(X,A)a_n\right\|_2\leq O\left(\tau_v\|\mathrm{out}^1_n(X,A)\|_2\left\|A\right\|_1\right)$ we can easily prove it.

5. Using (26) and (40), with larger enough $m$, we can prove it.

## C.3 Optimization

We write the gradient of loss function as $\nabla_{\mathbf{W}}\mathrm{Obj}_n(\mathbf{W})=\nabla_{\mathbf{W}}\mathrm{Obj}^1_n(\mathbf{W})+\nabla_{\mathbf{W}}\mathrm{Obj}^2_n(\mathbf{W})$, where $\nabla_{\mathbf{W}}\mathrm{Obj}^1_n(\mathbf{W})=\nabla_{\mathbf{W}}\mathrm{out}^1_n(X,A)$ and $\nabla_{\mathbf{W}}\mathrm{Obj}^2_n(\mathbf{W})=\nabla_{\mathbf{W}}CD^n_VV\mathrm{out}^1(X,A)a_n$, we can write its gradient as follows.

$$\begin{aligned}\left\langle\nabla_{\mathbf{W}}\mathrm{Obj}^1_n(\mathbf{W}),-\mathbf{W}'\right\rangle&=tr(Xa_n(y_n-\mathrm{out}_n(X,A))^\top CD^n_{W+W_0}\mathbf{W}')\\&=tr((y_n-\mathrm{out}_n(X,A))^\top CD^n_{W+W_0}W'Xa_n)\\&=\left\langle y_n-\mathrm{out}_n(X,A),CD^n_{W+W_0}W'Xa_n\right\rangle\end{aligned} \tag{44}$$

$$\begin{aligned}\left\langle\nabla_{\mathbf{W}}\mathrm{Obj}^2_n(\mathbf{W}),-\mathbf{W}'\right\rangle&=tr(\sum_{i=1}^N a_{ni}Xa_i(y_n-\mathrm{out}_n(X,A))^\top CD^n_{V+V_0}(\mathbf{V}^{(0)}+\mathbf{V})CD^i_{W+W_0}\mathbf{W}')\\&=tr(\sum_{i=1}^N a_{ni}(y_n-\mathrm{out}_n(X,A))^\top CD^n_{V+V_0}(\mathbf{V}^{(0)}+\mathbf{V})CD^i_{W+W_0}\mathbf{W}'Xa_i)\\&=tr((y_n-\mathrm{out}_n(X,A))^\top\sum_{i=1}^N a_{ni}CD^n_{V+V_0}(\mathbf{V}^{(0)}+\mathbf{V})CD^i_{W+W_0}\mathbf{W}'Xa_i)\\&=\left\langle y_n-\mathrm{out}_n(X,A),CD^n_{V+V_0}(\mathbf{V}^{(0)}+\mathbf{V})C(D_{W+W_0}\odot W'XA)a_n\right\rangle\end{aligned} \tag{45}$$

$$\begin{aligned}\left\langle\nabla_{\mathbf{V}}\mathrm{Obj}_n(\mathbf{V}),-\mathbf{V}'\right\rangle&=tr(\mathrm{out}(X)a_n(y_n-\mathrm{out}_n(X,A))^\top CD^n_{V+V_0}\mathbf{V}')\\&=tr((y_n-\mathrm{out}_n(X,A))^\top CD^n_{V+V_0}\mathbf{V}'\mathrm{out}(X)a_n)\\&=\left\langle y_n-\mathrm{out}_n(X,A),CD^n_{V+V_0}\mathbf{V}'\mathrm{out}(X)a_n\right\rangle\end{aligned} \tag{46}$$

Let us define $f(\mathbf{W}')=CD^n_{W+W_0}W'Xa_n+CD^n_{V+V_0}(\mathbf{V}^{(0)}+\mathbf{V})C(D_{W+W_0}\odot W'XA)a_n$ and $g(\mathbf{V}')=CD^n_{V+V_0}\mathbf{V}'\mathrm{out}(X)a_n$. Therefore,

$$\left\langle\nabla_{\mathbf{W},\mathbf{V}}\mathrm{Obj}_n(\mathbf{W},\mathbf{V}),(-\mathbf{W}',-\mathbf{V}')\right\rangle=\left\langle y_n-\mathrm{out}_n(X,A),f(\mathbf{W}')+g(\mathbf{V}')\right\rangle \tag{47}$$

**Claim C.4.** *We have that for all* $\mathbf{W}$ *and* $\mathbf{V}$ *satisfying* $\|\mathbf{W}\|_F \leq \tau_w$ *and* $\|\mathbf{V}\|_F \leq \tau_v$, *it holds that*

$$\|\nabla_{\mathbf{W}} \mathrm{Obj}(\mathbf{W}, \mathbf{V}; (x,y))\|_F \leq \|A\|_1 \|y_n - \mathrm{out}_n(X, A)\|_2 \cdot O(\sigma_v + 1) \tag{48}$$

$$\|\nabla_{\mathbf{V}} \mathrm{Obj}(\mathbf{W}, \mathbf{V}; (x,y))\|_F \leq \tau_w \|A\|_1 \|y_n - \mathrm{out}_n(X, A)\|_2 \cdot O(1) \tag{49}$$

Proof:

$$
\begin{aligned}
\|\nabla_{\mathbf{W}} \mathrm{Obj}(\mathbf{W}, \mathbf{V}; (x,y))\|_F &= \|X a_n (y_n - \mathrm{out}_n(X, A))^\top \\
&\quad \times (CD^n_{W+W_0} + CD^n_{V+V_0}(\mathbf{V}^{(0)} + \mathbf{V})CD^n_{W+W_0})\|_F \\
&\leq \|X a_n\|_2 \|y_n - \mathrm{out}_n(X, A)\|_2 \\
&\quad \times \|CD^n_{W+W_0} + CD^n_{V+V_0}(\mathbf{V}^{(0)} + \mathbf{V})CD^n_{W+W_0}\|_2 \\
&\leq \|A\|_1 \|y_n - \mathrm{out}_n(X, A)\|_2 \cdot O(\sigma_v + 1)
\end{aligned}
\tag{50}
$$

In (50), the last inequality uses $\|V^{(0)}\|_2 = O(\tau_v)$ and $\|C\|_2 \leq 1$.

$$
\begin{aligned}
\|\nabla_{\mathbf{V}} \mathrm{Obj}(\mathbf{W}, \mathbf{V}; (x,y))\|_F &= \|\mathrm{out}_n(X, A) a_n (y_n - \mathrm{out}_n(X, A))^\top CD^n_{V+V_0}\|_F \\
&\leq \|\mathrm{out}_n(X, A) a_n\|_2 \|y_n - \mathrm{out}_n(X, A)\|_2 \|CD^n_{V+V_0}\|_2 \\
&\leq \tau_w \|A\|_1 \|y_n - \mathrm{out}_n(X, A)\|_2 \cdot O(1)
\end{aligned}
\tag{51}
$$

In (51), the last inequality uses (27) and $\|C\|_2 \leq 1$.

**Claim C.5.** *In the setting of Lemma C.1, we have* $f(\mathbf{W}^* - \mathbf{W}) + g(\mathbf{V}^* - \mathbf{V}) = \mathcal{H}_{n,A^*}(X, A) - \mathrm{out}_n(X, A) + Err_n$ *with*

$$
\begin{aligned}
\underset{n \in \mathcal{V}, (X, y_n) \sim \mathcal{D}}{\mathbb{E}} \|Err_n\|_2 &\leq \underset{n \in \mathcal{V}, (X, y_n) \sim \mathcal{D}}{\mathbb{E}} [O(\tau_v \|A\|_1 + \alpha \mathcal{L}_{\mathcal{G}} \|A\|_1) \cdot \|\mathcal{H}_{n,A^*}(X, A) - \mathrm{out}_n(X, A)\|_2] \\
&\quad + O\left(\tau_v^2 \|A\|_1^2 \mathfrak{B}_{\mathcal{F}} + \tau_v \widetilde{\alpha}^2 \|A\|_1^2 + \alpha \tau_v \|A\|_1^2 \mathcal{L}_{\mathcal{G}} \mathfrak{B}_{\mathcal{F}}\right)
\end{aligned}
\tag{52}
$$

Proof: Based on the definition of $f(\mathbf{W}')$ and $g(\mathbf{V}')$, we have

$$
\begin{aligned}
f(\mathbf{W}^* - \mathbf{W}; X, a_n^*) + g(\mathbf{V}^* - \mathbf{V}; X, a_n^*) &= CD^n_{W+W_0}(W^* - W)X a_n \\
&\quad + CD^n_{V+V_0}(V^* - V)\mathrm{out}(X)a_n \\
&\quad + CD^n_{V+V_0}(\mathbf{V}^{(0)} + \mathbf{V})C(D_{W+W_0} \odot (W^* - W)XA)a_n \\
&= \underbrace{CD^n_{V+V_0}(\mathbf{V}^{(0)} + \mathbf{V})C(D_{W+W_0} \odot (W^* - W)XA)a_n}_{\clubsuit} \\
&\quad + \underbrace{CD^n_{W+W_0}W^* X a_n + CD^n_{V+V_0}V^* \mathrm{out}^1(X, A)a_n}_{\spadesuit} \\
&\quad - \underbrace{(CD^n_{W+W_0}WX a_n + CD^n_{V+V_0}V \mathrm{out}^1(X, A)a_n)}_{\diamondsuit}
\end{aligned}
\tag{53}
$$

1. For the $\clubsuit$ term,

$$
\begin{aligned}
\clubsuit &\leq \left(\left\|CD^n_{V+V_0}\mathbf{V}^{(0)}\right\|_2 + \|\mathbf{C}\|_2^2 \|\mathbf{V}\|_2\right) \|C(D_{W+W_0} \odot (W^* - W)XA)a_n\|_2 \\
&\leq O(1) \cdot O(\tau_v) \cdot \sum_{i=1}^N a_{ni} \left(\|\mathcal{F}(x) - \mathrm{out}_n^1(X, A)\|_2 + O(\widetilde{\alpha}^2 \|A\|_1)\right) \\
&\leq O(\tau_v) \left(\|\mathcal{F}(x) - \mathrm{out}_i(x)\|_2 \|A\|_1 + O(\widetilde{\alpha}^2 \|A\|_1^2)\right)
\end{aligned}
\tag{54}
$$

together with $\tau_v \leq \frac{1}{\mathrm{polylog}(m)} \sigma_v$.

2. For the ♠ term,

$$
\begin{aligned}
\spadesuit - (\mathcal{F}_n(X,A) + \alpha\mathcal{G}(\mathcal{F}(x), a_n) = {} & CD_{W+W_0}^n W^* X a_n - \mathcal{F}_n(X,A) \\
& + CD_{V+V_0}^n V^* \operatorname{out}^1(X,A) a_n - \alpha\mathcal{G}\left(\operatorname{out}_n^1(X,A), a_n\right) \\
& + \alpha\mathcal{G}\left(\operatorname{out}_n^1(X,A), a_n\right) - \alpha\mathcal{G}\left(\mathcal{F}(x), a_n\right)
\end{aligned}
\tag{55}
$$

The first term uses (38), the second term uses (39) and the third term uses the Lipscthiz continuity of $\mathcal{G}$, so we have

$$
\begin{aligned}
\|\spadesuit - (\mathcal{F}(x) + \alpha\mathcal{G}(\mathcal{F}(x)))\|_2 \leq {} & O\left(\widetilde{\alpha}^2 + \tau_v(\frac{\tau_v}{\sigma_v})^{1/3}\right) \cdot \|\operatorname{out}_n^1(X, a_n)\|_2 \|A\|_1 \\
& + O\left(\alpha\mathfrak{L}_\mathcal{G}\right) \|\mathcal{F}(X)a_n - \operatorname{out}_n^1(X)a_n\|_2 \\
\leq {} & O\left(\tau_v^2\right) \cdot \|\operatorname{out}_n^1(x)\|_2 \|A\|_1 + O\left(\alpha\mathfrak{L}_\mathcal{G}\right) \|\mathcal{F}_n(x) - \operatorname{out}_n^1(x)\|_2 \|A\|_1
\end{aligned}
\tag{56}
$$

We use $\frac{1}{\sigma_v} \leq \tau_v^2$ and definition of $\widetilde{\alpha}$.

3. For the ♢ term,

$$
\|\diamondsuit - \operatorname{out}_n(X)\|_2 \leq O\left(\left(\|\operatorname{out}_n^1(x)\|_2 \|A\|_1\right)\tau_v^2\right)
\tag{57}
$$

where the inequality uses (26) and (29).

In sum, we have

$$
Err \overset{\text{def}}{=} f\left(\mathbf{W}^* - \mathbf{W}; x\right) + g\left(\mathbf{V}^* - \mathbf{V}; x\right) - (\mathcal{F}(x) + \alpha\mathcal{G}(\mathcal{F}(x)) - \operatorname{out}_n(X,A)
\tag{58}
$$

satisfy

$$
\begin{aligned}
\underset{n\in\mathcal{V},(X,y_n)\sim\mathcal{D}}{\mathbb{E}} \|Err\|_2 \leq {} & \underset{n\in\mathcal{V},(X,y_n)\sim\mathcal{D}}{\mathbb{E}} \left[ O\left(\tau_v \|A\|_1 + \alpha\mathfrak{L}_\mathcal{G}\|A\|_1\right) \right. \\
& \left. \times \|\mathcal{F}(x) - \operatorname{out}_n^1(X,A)\|_2 + O\left(\|\operatorname{out}_n^1(x)\|_2 \|A\|_1\right)\tau_v^2 \right] \\
& + O\left(\tau_v\widetilde{\alpha}^2 \|A\|_1^2\right)
\end{aligned}
\tag{59}
$$

Using $\|\operatorname{out}_n^1(x)\|_2 \leq \|\operatorname{out}_n^1(X,A) - \mathcal{F}(x)\|_2 + \mathfrak{B}_\mathcal{F}$, we have

$$
\begin{aligned}
\underset{n\in\mathcal{V},(X,y_n)\sim\mathcal{D}}{\mathbb{E}} \|Err\|_2 \leq {} & \underset{n\in\mathcal{V},(X,y_n)\sim\mathcal{D}}{\mathbb{E}} \left[O\left(\tau_v \|A\|_1 + \alpha\mathfrak{L}_\mathcal{G}\|A\|_1\right) \cdot \|\mathcal{H}(x) - \operatorname{out}_n(X,A)\|_2\right] \\
& + O\left(\tau_v^2 \|A\|_1 \mathfrak{B}_\mathcal{F} + \tau_v\widetilde{\alpha}^2 \|A\|_1^2\right) \\
& + O\left(\tau_v \|A\|_1 + \alpha\mathfrak{L}_\mathcal{G}\|A\|_1\right) \cdot \left(\tau_v \|A\|_1 \mathfrak{B}_\mathcal{F} + \alpha\mathfrak{B}_{\mathcal{F}\circ\mathcal{G}}\right)
\end{aligned}
\tag{60}
$$

Using $\mathfrak{B}_{\mathcal{F}\circ\mathcal{G}} \leq \sqrt{k}p_\mathcal{G}\mathfrak{C}_\mathfrak{s}(\mathcal{G}, \|A\|_1 \mathfrak{B}_\mathcal{F})(\|A\|_1 \mathfrak{B}_\mathcal{F})^2\mathfrak{B}_\mathcal{F} \leq \frac{\tau_v}{\alpha}\mathfrak{B}_\mathcal{F}$, so we have

$$
\begin{aligned}
\underset{n\in\mathcal{V},(X,y_n)\sim\mathcal{D}}{\mathbb{E}} \|Err\|_2 \leq {} & \underset{n\in\mathcal{V},(X,y_n)\sim\mathcal{D}}{\mathbb{E}} \left[O\left(\tau_v \|A\|_1 + \alpha\mathfrak{L}_\mathcal{G}\|A\|_1\right) \cdot \|\mathcal{H}(x) - \operatorname{out}_n(X,A)\|_2\right] \\
& + O\left(\tau_v^2 \|A\|_1^2 \mathfrak{B}_\mathcal{F} + \tau_v\widetilde{\alpha}^2 \|A\|_1^2 + \alpha\tau_v \|A\|_1^2 \mathfrak{L}_\mathcal{G}\mathfrak{B}_\mathcal{F}\right)
\end{aligned}
\tag{61}
$$

**Claim C.6.** *In the setting of Lemma C.1, if $\tau_v \|A\|_1 \leq \frac{1}{polylog(m)}$, we have*

$$
\left\|\operatorname{out}_n^1(X,A) - \mathcal{F}_n(X)\right\|_2 \leq 2\left\|\operatorname{out}_n^1(X,A) - \mathcal{H}_n(X,A)\right\|_2 + \alpha\mathfrak{B}_{\mathcal{F}\circ\mathcal{G}} + \widetilde{O}\left(\tau_v \|A\|_1 \mathfrak{B}_\mathcal{F}\right)
\tag{62}
$$

Proof: Using 31 and $\|\mathcal{G}(\mathcal{F}(X), a_n)\|_2 \leq \mathfrak{B}_{\mathcal{F}\circ\mathcal{G}}$, we have

$$
\begin{aligned}
\left\|\operatorname{out}_n^1(X,A) - \mathcal{F}_n(X)\right\|_2 \leq {} & \left\|\operatorname{out}_n^1(X,A) - \mathcal{H}_n(X,A)\right\|_2 + \alpha\mathfrak{B}_{\mathcal{F}\circ\mathcal{G}} \\
& + \widetilde{O}\left(\tau_v \|A\|_1 \left(\|\operatorname{out}_n^1(X, a_n) - \mathcal{F}_n(X,A)\|_2 + \mathfrak{B}_\mathcal{F}\right)\right)
\end{aligned}
\tag{63}
$$

Using $\tau_v \|A\|_1$ small enough, we have

$$
\left\|\operatorname{out}_n^1(X,A) - \mathcal{F}_n(X)\right\|_2 \leq 2\left\|\operatorname{out}_n^1(X,A) - \mathcal{H}_n(X,A)\right\|_2 + \alpha\mathfrak{B}_{\mathcal{F}\circ\mathcal{G}} + \widetilde{O}\left(\tau_v \|A\|_1 \mathfrak{B}_\mathcal{F}\right)
\tag{64}
$$

### C.4 Proof of Theorem 3.2

Proof: Using (47) and Claim C.5, in iteration $t$, we have

$$
\begin{aligned}
&\langle \nabla_{\mathbf{W},\mathbf{V}}\, \mathrm{Obj}_n\left(\mathbf{W}_t, \mathbf{V}_t\right)), \left(\mathbf{W}_t - \mathbf{W}^*, \mathbf{V}_t - \mathbf{V}^*\right))\rangle \\
&= \langle y_n - \mathrm{out}\left(\mathbf{W}_t, \mathbf{V}_t\right), f\left(\mathbf{W}^* - \mathbf{W}\right) + g\left(\mathbf{V}^* - \mathbf{V}\right)\rangle \\
&= \langle y_n - \mathrm{out}_n\left(\mathbf{W}_t, \mathbf{V}_t\right), \mathcal{H}_{n,A^*}(X) - \mathrm{out}_n\left(\mathbf{W}_t, \mathbf{V}_t\right) + Err_t\rangle
\end{aligned}
\tag{65}
$$

We also have

$$
\begin{aligned}
\|W_{t+1} - W^*\|_F^2 &= \|W_t - \eta_w \nabla_W\, \mathrm{Obj}_n(W_t) - W^*\|_F^2 \\
&= \|W_t - W^*\|_F^2 - 2\eta_w \langle \nabla_W\, \mathrm{Obj}_n(W_t), W_t - W^*\rangle \\
&\quad + \eta_w^2 \|\nabla_W\, \mathrm{Obj}_n(W_t)\|_F^2, \\
\|V_{t+1} - V^*\|_F^2 &= \|V_t - \eta_v \nabla_V\, \mathrm{Obj}_n(V_t) - V^*\|_F^2 \\
&= \|V_t - V^*\|_F^2 - 2\eta_v \langle \nabla_V\, \mathrm{Obj}_n(V_t), V_t - V^*\rangle \\
&\quad + \eta_v^2 \|\nabla_V\, \mathrm{Obj}_n(V_t)\|_F^2
\end{aligned}
\tag{66}
\tag{67}
$$

Using Algorithm 1, we have $\mathbf{W}_{t+1} = \mathbf{W}_t - \eta_w \nabla_{\mathbf{W}}\, \mathrm{Obj}_n\left(\mathbf{W}_t, \mathbf{V}_t\right)$ and $\mathbf{V}_{t+1} = \mathbf{V}_t - \eta_v \nabla_{\mathbf{V}}\, \mathrm{Obj}_n\left(\mathbf{W}_t, \mathbf{V}_t\right)$, so we have

$$
\begin{aligned}
&\langle \nabla_{\mathbf{W},\mathbf{V}}\, \mathrm{Obj}_t\left(\mathbf{W}_t, \mathbf{V}_t\right), \left(\mathbf{W} - \mathbf{W}^*, \mathbf{V} - \mathbf{V}^*\right))\rangle \\
&= \underbrace{\frac{\eta_w}{2}\|\nabla_{\mathbf{W}}\, \mathrm{Obj}_t\left(\mathbf{W}_t, \mathbf{V}_t\right)\|_F^2 + \frac{\eta_v}{2}\|\nabla_{\mathbf{V}}\, \mathrm{Obj}_t\left(\mathbf{W}_t, \mathbf{V}_t\right)\|_F^2}_{\heartsuit} \\
&\quad + \frac{1}{2\eta_w}\|\mathbf{W}_t - \mathbf{W}^*\|_F^2 - \frac{1}{2\eta_w}\|\mathbf{W}_{t+1} - \mathbf{W}^*\|_F^2 + \frac{1}{2\eta_v}\|\mathbf{V}_t - \mathbf{V}^*\|_F^2 - \frac{1}{2\eta_v}\|\mathbf{V}_{t+1} - \mathbf{V}^*\|_F^2
\end{aligned}
\tag{68}
$$

Using Claim C.4 and change all $A$ to $A^*$, we have

$$
\begin{aligned}
\heartsuit &\leq O\left(\eta_w \sigma_v^2 + \eta_v \tau_w^2\right) \cdot \|A\|_1^2 \|y_n - \mathrm{out}_n(X, A^*)\|_2^2 \\
&\leq O\left(\eta_w \sigma_v^2 + \eta_v \tau_w^2\right) \cdot \|A\|_1^2 \left(\|\mathcal{H}_{n,A^*}(X) - \mathrm{out}_n(X, A^*)\|_2^2 + \|\mathcal{H}_{n,A^*}(X) - y_n\|_2^2\right)
\end{aligned}
\tag{69}
$$

Therefore, as long as $O\left(\eta_w \sigma_v^2 + \eta_v \tau_w^2\right) \leq 0.1$, it satisfies

$$
\begin{aligned}
\frac{1}{4}\|\mathcal{H}_{n,A^*}(X) - \mathrm{out}_n(X, A^*)\|_2^2 \leq\, & 2\|\mathrm{Err}_t\|_2^2 + 4\|\mathcal{H}_{n,A^*}(X) - y_n\|_2^2 \\
& + \frac{1}{2\eta_w}\|\mathbf{W}_t - \mathbf{W}^*\|_F^2 - \frac{1}{2\eta_w}\|\mathbf{W}_{t+1} - \mathbf{W}^*\|_F^2 \\
& + \frac{1}{2\eta_v}\|\mathbf{V}_t - \mathbf{V}^*\|_F^2 - \frac{1}{2\eta_v}\|\mathbf{V}_{t+1} - \mathbf{V}^*\|_F^2
\end{aligned}
\tag{70}
$$

After telescoping for $t = 0, 1, \ldots, T_0 - 1$,

$$
\begin{aligned}
&\frac{\|\mathbf{W}_{T_0} - \mathbf{W}^*\|_F^2}{2\eta_w T_0} + \frac{\|\mathbf{W}_{T_0} - \mathbf{V}^*\|_F^2}{2\eta_v T_0} + \frac{1}{2T_0}\sum_{t=0}^{T_0-1}\|\mathcal{H}_{n,A^*}(X) - \mathrm{out}_n(X, A^*)\|_2^2 \\
&\leq \frac{\|\mathbf{W}^*\|_F^2}{2\eta_w T_0} + \frac{\|\mathbf{V}^*\|_F^2}{2\eta_v T_0} + \frac{O(1)}{T_0}\sum_{t=0}^{T_0-1}\|Err_t\|_2^2 + \|\mathcal{H}_{n,A^*}(X) - y_t\|_2^2.
\end{aligned}
\tag{71}
$$

Using $O\left(\tau_v \|A\|_1 + \alpha \mathfrak{L}_{\mathcal{G}}\right) \leq 0.1$, we have

$$
\frac{1}{4T}\sum_{t=0}^{T-1} \mathbb{E}_{n \in \mathcal{V}, (X, y_n) \sim \mathcal{Z}}\|\mathcal{H}_{n,A^*}(X) - \mathrm{out}_n(X, A^*)\|_2^2 \leq \frac{\|\mathbf{W}^*\|_F^2}{2\eta_w T} + \frac{\|\mathbf{V}^*\|_F^2}{2\eta_v T} + O\left(\mathrm{OPT} + \epsilon_0\right)
\tag{72}
$$

where

$$
\begin{aligned}
\epsilon_0 &= \Theta\left(\widetilde{\alpha}^2\tau_v\left\|A^*\right\|_1^2 + \tau_v^2\left\|A^*\right\|_1\mathfrak{B}_{\mathcal{F}} + \alpha\tau_v\left\|A^*\right\|_1\mathfrak{L}_{\mathcal{G}}\mathfrak{B}_{\mathcal{F}}\right)^2 \\
&= \widetilde{\Theta}\left(\widetilde{\alpha}^2\tau_v\left\|A^*\right\|_1^2 + \alpha^2(kp_{\mathcal{G}}\mathfrak{C}_{\mathfrak{s}}(\mathcal{G},\mathfrak{B}_{\mathcal{F}}\left\|A^*\right\|_1)^2(\mathfrak{B}_{\mathcal{F}}\left\|A^*\right\|_1)^3 \right. \\
&\quad\left. + \alpha^2 kp_{\mathcal{G}}\mathfrak{C}_{\mathfrak{s}}(\mathcal{G},\mathfrak{B}_{\mathcal{F}}\left\|A^*\right\|_1)(\mathfrak{