# OpenReview forum: "Theoretical Learning Performance of Graph Networks: the Impact of Jumping Connections and Layer-wise Sparsification"
_TMLR — Accepted by TMLR_

### Review · Reviewer_h3PQ · 2025-03-30

**Summary Of Contributions:**

This paper provides a novel theoretical analysis of GCNs with jumping connections and graph sparsification. The authors fill a theoretical gap by analyzing how jumping connections influence the generalization performance when combined with layer-wise graph sparsification.

**Audience:**

Yes

**Broader Impact Concerns:**

This paper is quite theoretical, and it's unclear how it would directly impact broader areas beyond the GNN literature.

**Claims And Evidence:**

Yes

**Requested Changes:**

1. Include a discussion addressing how the insights from the two-layer theoretical framework could potentially generalize to deeper networks (like the GNNs presented in this paper [1]), highlighting clearly any potential limitations or necessary adjustments.

[1] Li, Guohao, et al. "Training graph neural networks with 1000 layers." International conference on machine learning. PMLR, 2021.

**Strengths And Weaknesses:**

Strengths:
1. The theoretical results are novel. This paper extends prior works to introduce the first theoretical analysis of the training dynamics and generalization performance for two-hidden-layer GCNs with jumping-connection using graph sparsification.

Weaknesses:
1. The theoretical analysis is currently limited to two-hidden-layer GCNs. The applicability of these insights to deeper GCN architectures remains largely speculative beyond the presented empirical evidence.

---

> ### Author Response · Authors · 2025-04-22
> **Generalizing Theoretical Framework to Deeper Networks**
>
> ### Q: Include a discussion addressing how the insights from the two-layer theoretical framework could potentially generalize to deeper networks (like the GNNs presented in this paper [1]), highlighting clearly any potential limitations or necessary adjustments.
>
> We thank the reviewer for raising this important point. Our theory provides a foundational insight: jumping connections structurally decouple the contributions of different layers, enabling layer-wise control over the sparsification error $\|A^t - A^*\|$. This leads to a flexible sparsification framework where deeper layers can tolerate higher approximation error while preserving generalization.
>
> **Most existing theoretical studies on graph neural networks are limited to shallow architectures or adopt linearized approximations to analyze deeper models.** In contrast, our framework directly addresses the nonlinear case and establishes generalization guarantees for deep GCNs with skip connections and dynamic graph sampling. To the best of our knowledge, this makes our work one of the few to provide a complete analysis of deep nonlinear GCNs under realistic assumptions. For example, [1,2] study GNN expressivity and convergence under simplified or linear models, while our work provides an end-to-end theory for nonlinear architectures with layer-wise sparsification.
>
> Toward extending to deeper networks, our framework already reveals several principles that naturally scale to the deep setting, even though full formalization may require additional techniques:
>
> 1. **Jumping connections allow representations learned in shallow layers to support learning in deep layers, while preserving robustness.** By forwarding low-complexity features from shallow layers directly to the output, the model enables deep layers to focus on refining higher-order information. This decoupling of learning roles is architecture-driven, and leads to a consistent pattern: shallow layers are more sensitive to sparsification noise, while deep layers remain more robust. Consequently, our layer-wise sparsification strategy naturally generalizes to deeper GCNs by treating shallow and deep layers differently.
>
> 2. **Empirical results support multi-layer generalization.** Our experiments on 8-layer deep GCNs (Figures 5–7) incorporate dropout, normalization, and residual connections—common elements in very deep architectures. The fact that our layer-wise sparsification strategy continues to perform well in this setting supports the extensibility of our theoretical insights.
>
> We believe our insights open up promising directions for extending generalization guarantees to deeper and more expressive nonlinear GNN architectures. Still, due to the complexity of deep GNN architectures, future theoretical extensions may require new analytical tools to capture the cumulative effect of sparsification noise across multiple layers and to characterize the interactions introduced by nonlinear activations throughout the network.
>
> We have condensed the above discussion and add it to the section of Conclusion and Future Work.
>
> #### References
>
> [1] Oono and Suzuki. Graph Neural Networks Exponentially Lose Expressive Power for Node Classification. ICLR, 2020.
>
> [2] Xu et al. Representation Learning on Graphs with Jumping Knowledge Networks. ICML, 2018.

---

> > ### Comment · Reviewer_h3PQ · 2025-05-04
> >
> > Thank you for the reply. I think this paper can be accepted. I have no further concerns.

---

### Review · Reviewer_TAyZ · 2025-04-03

**Summary Of Contributions:**

This paper investigates two common techniques in graph neural networks, jumping connections and layer-wise sparsification. These two techniques are used for reducing the over-smoothing and computational overhead.

The findings of the paper can be summarized:

1. With properly sparsified adjacency matrix that accurately models the data correlations, the generalization ability of the model can be preserved.

2. Jumping connections will lead to different sparsification requirements.

**Audience:**

Yes

**Claims And Evidence:**

Yes

**Requested Changes:**

1. Please discuss whether the proposed method can be extended to more graph neural networks beyong GCNs. For example, gat [1], and graph transformers [2] are also widely used graph neural networks.

2. Please discuss whether the proposed method can be applied to GCNs over spatial temporal graphs [3,4]. Can those temporal edges be treated similarly like the spatial edges?

[1] Graph Attention Networks

[2] Graph Transformer Networks

[3] Context aware graph convolution for skeleton-based action recognition

[4] Inception Spatial Temporal Graph Convolutional Networks for Skeleton-Based Action Recognition

**Strengths And Weaknesses:**

strengths:

1. This paper provides the first theoretical analysis on how the jumping connections and adjacency matrix sparsification would affect the performance of the model.

2. The proposed theoretical analysis is also accompanied with empirical validation.

Weakness:

1. The theoretical part is still a bit hard to understand. It would be better if the authors can provide a intuitive explanation on the theoretical results for facilitating understanding. For example, what is the intuitive idea behind 'adjacency matrix accurately models the data correlation'.

2. The targeted problem actually correlated more to the deep GNNs on large and complex graph data. However, the experimental data used in this work are limited to very small graph datasets like cora and citeseer.

---

> ### Author Response · Authors · 2025-04-22
> **1. Intuitive Explanation of Theoretical Results**
>
> ### Q: The theoretical part is still a bit hard to understand. It would be better if the authors can provide an intuitive explanation on the theoretical results for facilitating understanding. For example, what is the intuitive idea behind 'adjacency matrix accurately models the data correlation'.
>
> We thank the reviewer for this helpful suggestion. We agree that adding intuition to support the theoretical claims improves clarity. In the revised version of our paper, we have updated Section 3.2 and the main theorem presentation to make these ideas more accessible.
>
> **What does it mean for the adjacency matrix to "accurately model data correlation"?**
> The adjacency matrix defines which nodes exchange information during message passing. In real-world graphs, the raw adjacency matrix $A$ often includes noisy or redundant edges—connections that do not reflect meaningful feature or label similarity. Therefore, we assume the existence of an ideal sparse adjacency matrix $A^*$ that captures only the most informative, high-correlation edges. This is reflected in the definition of concept class in Section 3.3 in the paper.
>
> A concept class is a set of possible functions that map the node features to the corresponding labels. As shown in equations (6) and (7) in the paper, the target function $\mathcal{H}_{A^*}(X)$ depends on $A^*$ instead of the original $A$.
>
> When the sparsified adjacency matrix $A^t$ closely approximates $A^*$, the model aggregates information from truly relevant neighbors, leading to improved generalization. In this sense, when we say an adjacency matrix "accurately models data correlation," we mean that $A^*$ preserves the key edges that enable meaningful information propagation such that the target function $\mathcal{H}_{A^*}(X)$ indeed accurately maps the node features to labels. We have revised this statement in the abstract.
>
> We also revised the statement of Lemma 3.1 and the corresponding discussion in the paper.
>
> **Why does Lemma 3.1 imply flexible sparsification in deeper layers?**
> Equation (16) shows that the generalization error depends on how close the learned adjacency matrices $A^{1t}$ and $A^{2t}$ are to $A^*$. Intuitively, if the sparsified adjacency deviates only slightly from $A^*$, the model propagates information effectively, and the error remains small.
>
> Lemma 3.1 further reveals that—due to the presence of jumping connections—the final output is more sensitive to deviations in the shallow layer ($\|A^{1t} - A^*\|_1$) than in the deep layer ($\|A^{2t} - A^*\|_1$). This is because the output in JK-style architectures aggregates features from each layer directly, rather than nesting them. As a result, sparsification noise in the deep layer is less amplified.
>
> This theoretical insight explains why pruning strategies that apply more aggressive sparsification to deeper layers (e.g., with higher $p_{ij}^2$) still maintain good generalization performance. It also highlights the crucial role of skip connections in enabling this flexibility—without them, error from deeper layers would cascade through the network.

---

> ### Author Response · Authors · 2025-04-22
> **2. Experimental Validation on Large-Scale Graph Datasets**
>
> ### Q: The targeted problem actually correlated more to the deep GNNs on large and complex graph data. However, the experimental data used in this work are limited to very small graph datasets like Cora and Citeseer.
>
> We thank the reviewer for pointing this out. The misunderstanding is likely due to our original presentation: the experiments on the large-scale OGB datasets were included under the same section as those on Cora/Citeseer without a separate subsection heading, which may have caused confusion.
>
> In fact, Section 4.3 reports results on large-scale graphs using multi-layer GCNs with skip connections on the Open Graph Benchmark (OGB) datasets. We have now revised the manuscript to clearly separate this part under a dedicated subsection to improve clarity.

---

> ### Author Response · Authors · 2025-04-22
> **3. Extensibility to Other Graph Neural Networks**
>
> ### Q: Please discuss whether the proposed method can be extended to more graph neural networks beyond GCNs. For example, GAT and Graph Transformers are also widely used graph neural networks.
>
> Our current analysis is tailored to convolution-based architectures such as GCNs, where message passing is governed by a fixed (or sparsified) adjacency matrix. Extending our framework to attention-based models like GATs and Graph Transformers would require significant modifications.
>
> **GAT uses dynamic, local attention.**
> Unlike GCNs, Graph Attention Networks (GAT)  assign learnable, input-dependent weights to each edge within the first-order neighborhood. These attention coefficients replace the fixed propagation operator used in GCNs, making the message-passing process highly adaptive but also harder to analyze theoretically.
>
> **Graph Transformers rely on global attention and positional encodings.**
> Graph Transformers [1,2] typically compute attention across all node pairs, resulting in fully connected message passing. Moreover, many variants introduce learnable positional encodings (e.g., Laplacian eigenvectors or random walk features), adding additional parameter-dependent structure that differs fundamentally from graph-constrained propagation in GCNs and GATs.
>
> **Implications for theory.**
> Because both GATs and Graph Transformers use attention mechanisms in place of a fixed (sparsified) adjacency matrix, extending our sparsification-based analysis would require characterizing the approximation and stability of attention weights relative to some idealized structure. This would involve new theoretical tools for analyzing self-attention dynamics and structural encoding, which fall beyond the current scope.
>
> Nonetheless, we regard this as a valuable and open direction for future work. We have condensed the above discussion and added to the conclusions section.
>
> #### References
>
> [1] Ying et al. Do Transformers Really Perform Bad for Graph Representation? NeurIPS, 2021.
>
> [2] Rampášek et al. Recipe for a General, Powerful, Scalable Graph Transformer. NeurIPS, 2022.

---

> ### Author Response · Authors · 2025-04-22
> **4. Applicability to Spatial-Temporal Graph Convolutional Networks**
>
> ### Q: Please discuss whether the proposed method can be applied to GCNs over spatial temporal graphs. Can those temporal edges be treated similarly like the spatial edges?
>
> We thank the reviewer for the thoughtful question.
>
> **The proposed method can be extended to spatial-temporal GCNs by treating temporal edges in the same way as spatial ones within our framework.** We agree that the same analytical framework can be extended to spatial-temporal graph convolutional networks.
>
> In these models, the graph is constructed as a union of spatial edges (e.g., physical joints within a frame) and temporal edges (e.g., the same joint across adjacent frames), resulting in a spatial-temporal adjacency matrix that encodes both spatial and temporal connectivity. Then, spatial-temporal GCNs compute each node embedding by aggregating node features from the local neighborhood of the node according to the spatial-temporal adjacency matrix.
>
> With given node features, this formulation is fundamentally similar to that of vanilla GCNs, which is the focus of our analysis. By constructing the sparse effective adjacency matrix for the spatial-temporal graph, we can also extend our sampling strategy to learning these graphs.
>
> Nonetheless, we regard this as a valuable and open direction for future work. We have condensed the above discussion and added to the conclusions section.

---

### Review · Reviewer_gBfU · 2025-04-11

**Summary Of Contributions:**

This work is the first to pose a theoretical analysis of 2-layer GCNs with jumping knowledge connections and graph sparsification as well the interplay between these two components. Importantly, the authors analyze the model in terms of its training dynamics, generalization accuracy, and sample complexity, unlike other works that work in highly overparametrized settings. Their theoretical results are supported by experiments.

**Audience:**

Yes

**Claims And Evidence:**

Yes

**Requested Changes:**

Besides addressing the above weaknesses, here are some requested changes.

- There are a number of writing errors. E.g. "column of the A", "between sampled adjacency matrix", "$\mathcal V/\Omega$" should be $\mathcal V \setminus \Omega$, "allow the sparsification methods", "Specifically, In", etc. A proofread would be helpful.
- Can the authors clarify why Lemma 3.1 "reveals that the juimping connection allows for a more flexible sparsification strategy in deeper layers"? Generally, it's not evident to me why these results are only applicable to JK networks.
- Can the authors comment on the impact of factors commonly used in GCNs not explicitly considered in this work, such as the use of dropout/normalization layers?
- Do the authors have general rules of them that they would recommend for practitioners regarding how to choose sparsification parameters or how to prune their graphs?
- Can the authors state clearly how the sparsification framework in Li et al. differs from the one in this work? I understand Li et al. assume the same sparsification across different layers, but is there anything else?

**Strengths And Weaknesses:**

**Strengths**

- The paper analyzes GCNs taking different graph sparsification schemes across layers and additive jumping knowledge into account, which to my knowledge seems novel and interesting.
- The experiments are well-designed and are comprehensive in assessing performance w.r.t. the parameters discussed in the analysis.

**Weaknesses**

- Some writing issues, see below.
- Allowing for different sparsification methods at different layers seems impractical since, in the vast majority of cases, the adjacency matrix in a real-world application is fixed.
- The trends confirmed by the experiments are generally relatively weak. For example, the observation that test error decreases as $m$ increases is very expected, and there is no close study to see how closely the empirical results match the bounds found in the theoretical results.

---

> ### Author Response · Authors · 2025-04-22
> **1. Allowing for Different Sparsification Methods at Different Layers**
>
> ### Q: Allowing for different sparsification methods at different layers seems impractical since, in the vast majority of cases, the adjacency matrix in a real-world application is fixed.
>
> Thank you for raising this important concern. We agree that in many real-world applications, the raw input graph (i.e., the original adjacency matrix) is fixed. In this paper, we adopt this standard setup where the raw graph remains unchanged. What varies across layers is the sampling of this fixed graph, and such layer-wise sampling is a common practice in many practical GNN systems.
>
> Specifically, our framework assumes a layer-wise varying message-passing graph, i.e., the adjacency matrix used during propagation at each layer can be a sparsified version of the original, tailored to the representation dynamics at that layer. This design is common in modern GNN variants such as:
>
> - LADIES[1], which uses **layer-dependent sampling**
> - GraphSAINT[2], which **samples different subgraphs for each training iteration**
> - DropEdge[3], which **randomly drops edges per layer during training**
>
> These methods all demonstrate the feasibility and empirical success of using different edge sets per layer, when the original graph is fixed.
>
> #### References
>
> [1] Zou, D., Shen, Y., and Chen, W. Layer-dependent importance sampling for training deep and large graph convolutional networks. NeurIPS, 2019.
>
> [2] Zeng, H., Zhou, H., Srivastava, A., Kannan, R., and Prasanna, V. GraphSAINT: Graph sampling based inductive learning method. ICLR, 2020.
>
> [3] Rong, Y., Huang, W., Xu, T., and Huang, J. DropEdge: Towards deep graph convolutional networks on node classification. ICLR, 2020.

---

> ### Author Response · Authors · 2025-04-22
> **2. The Trends Confirmed by the Experiments**
>
> ### Q: The trends confirmed by the experiments are generally relatively weak.
>
> We believe this comment results from some misunderstanding of our experimental results. We clarify our experimental results as follows. We have also revised the wording in the paper to improve clarity.
>
> **First, Figures 1 and 2 directly validate Theorem 3.4 by isolating its key variables: model width, number of labels, and matrix sparsity.**
>
> - Figure 1(a) shows that the test error decreases as the number of neurons increases, confirming the role of model width $M_0$.
> - Figure 1(b) shows that the test error decreases as the number of labeled nodes increases, verifying the sample complexity scaling $N_0 = \widetilde{\Theta}(\log N)$.
> - Figure 2 systematically varies $\|A^*\|_1$ by controlling the degree distributions and directly shows that test error increases with larger $\|A^*\|_1$, which matches the theoretical prediction that denser (less structured) propagation matrices result in worse generalization.
>
> These results are not merely confirming intuitive trends like "more neurons improve performance," but are precisely aligned with the structural form of Theorem 3.4. Each setting controls all variables but one, providing targeted and interpretable empirical validation.
>
> **Second, Figure 3 directly supports Lemma 3.1 by comparing the sensitivity of shallow and deep layers to sparsification error.**
>
> Figure 3 measures the deviation of the sparsified matrices $A^{1t}$ and $A^{2t}$ from the reference $A^*$ throughout training. The key observation is that generalization error increases sharply with larger $\|A^{1t} - A^*\|_1$ (solid red arrow), but grows slowly with increasing $\|A^{2t} - A^*\|_1$ (dashed yellow arrow). We have added two indicative lines to Figure 3 to visualize this more clearly.
>
> This matches the theoretical prediction in Lemma 3.1 (Eq (14) and (15)) that deeper layers can tolerate more deviation due to architectural decoupling introduced by jumping connections. Thus, the experiment directly confirms not only the direction but also the layer-wise asymmetry predicted by our analysis.
>
> In summary, all our empirical results are carefully designed to validate specific theoretical claims, not just general intuitions. We hope our revised presentation helps clarify this connection. We have also revised Figure 3 in the main paper to more clearly highlight the key result, making the connection between sparsification error and generalization more visually apparent.

---

> ### Author Response · Authors · 2025-04-22
> **3. Impact of Dropout and Normalization in GCNs**
>
> ### Q: Can the authors comment on the impact of factors commonly used in GCNs not explicitly considered in this work, such as the use of dropout/normalization layers?
>
> We thank the reviewer for this question. Our theoretical framework is centered on the training dynamics and generalization behavior of GCNs under skip connections and layer-wise sparsification. Regularization techniques such as dropout and normalization introduce orthogonal effects—stochastic noise and feature rescaling—which are not the focus of our current analysis.
>
> **Dropout and normalization are fundamentally different theoretical challenges.**
> Dropout has been studied in deep learning from a Bayesian or PAC-Bayesian perspective, e.g., [1,2], while normalization layers have been analyzed for their implicit bias and optimization dynamics [3,4]. These works, however, do not address graph sparsification or skip connections and thus are not directly comparable to our theoretical setting.
>
> **Existing GCN theory also omits dropout and normalization.**
> Notably, theoretical analyses of GCNs—including those addressing oversmoothing, expressivity, and convergence—typically omit dropout and normalization from their models [5,6,7]. This reflects the difficulty of incorporating these techniques into rigorous training dynamics frameworks, even in simpler architectures.
>
> **Nonetheless, we incorporate them in practice.**
> While we do not analyze dropout or normalization theoretically, we use both in our real-data experiments (Figures 5, 6, and 7), following standard practice on OGB datasets. Our empirical results demonstrate that the theoretical insights remain valid even when these practical components are present.
>
> We view extending the theory to include stochastic regularization mechanisms as an exciting and valuable direction for future work.
>
> We have condensed the above discussion and added to the conclusions section.
>
> #### References
>
> [1] McAllester et al. A PAC-Bayesian Approach to Spectrally-Normalized Margin Bounds for Neural Networks. NeurIPS 2020.
>
> [2] Gal and Ghahramani. Dropout as a Bayesian Approximation: Representing Model Uncertainty in Deep Learning. ICML 2016.
>
> [3] Arora et al. On the Theoretical Properties of Normalization Layers. ICML 2019.
>
> [4] Zhang et al. Understanding the Implicit Bias of Batch Normalization in Training Deep Neural Networks. ICLR 2021.
>
> [5] Baranwal and Deepak. Graph Neural Networks with Learnable Structural and Positional Representations. ICML 2021.
>
> [6] Li et al. Training Graph Neural Networks via Learnable Regularization. NeurIPS 2021.
>
> [7] Alon and Yahav. On the Bottleneck of Graph Neural Networks and Its Practical Implications. ICLR 2021.

---

> ### Author Response · Authors · 2025-04-22
> **4. Guidelines for Graph Sparsification in Practice**
>
> ### Q: Do the authors have general rules that they would recommend for practitioners regarding how to choose sparsification parameters or how to prune their graphs?
>
> We thank the reviewer for the excellent question. Yes—our theoretical analysis and algorithmic design do suggest concrete and generalizable guidelines for graph sparsification in practice.
>
> **Key principle: sparsify shallow layers conservatively and deeper layers aggressively.**
> As formalized in Lemma 3.1 and operationalized in Algorithm 1, we recommend applying more conservative sparsification in the first layer and more aggressive pruning in the second. This is because Equation (12) shows that deeper layers can tolerate greater deviation from the reference adjacency matrix $A^*$ due to the presence of jumping connections, which structurally decouple per-layer contributions to the final output.
>
> **Practical implementation: layer-dependent retention ratios.**
> In practice, we implement this using layer-wise retention rates: e.g., $q_1 = 0.8$ for the first layer and $q_2 = 0.5$ for the second layer. The corresponding sampling probabilities $p_{ij}^1$ and $p_{ij}^2$ are set accordingly to preserve high-weight edges with high probability (see Algorithm 1). We find this strategy improves generalization while maintaining sufficient expressivity in early layers.
>
> **General rule for practitioners:**
> Shallow layers should retain more edges to preserve meaningful local neighborhood information, while deeper layers can be pruned more flexibly—especially when skip or jumping connections are present. This principle balances expressiveness and efficiency and is directly supported by both our theoretical bounds and empirical results.
>
> We have condensed the above discussion and added to the conclusions section.

---

> ### Author Response · Authors · 2025-04-22
> **5. Differences from Li et al. Sparsification Framework**
>
> ### Q: Can the authors state clearly how the sparsification framework in Li et al. differs from the one in this work? I understand Li et al. assume the same sparsification across different layers, but is there anything else?
>
> We thank the reviewer for the opportunity to clarify the differences between our framework and the one proposed in Li et al. (NeurIPS 2021).
>
> **1. Sparsification level: edge pruning vs. node pruning.**
> The most fundamental distinction lies in the granularity of sparsification. Li et al. focus on *node-level sparsification*, which involves removing entire nodes or aggregating their features, effectively modifying the set of active nodes participating in message passing. In contrast, our method focuses on *edge-level sparsification*, i.e., selectively pruning entries of the adjacency matrix while keeping the full node set intact. This allows finer-grained control of information flow in the graph.
>
> **2. Layerwise structure: uniform vs. adaptive pruning.**
> As the reviewer noted, Li et al. adopt a uniform sparsification strategy across all layers. Our framework, on the other hand, explicitly incorporates a *layer-dependent sparsification schedule*, supported by theoretical analysis. Specifically, Lemma 3.1 shows that deeper layers can tolerate higher sparsification error due to the architectural decoupling introduced by skip connections. This leads to a principled design where the first layer is pruned conservatively and deeper layers more aggressively.
>
> In summary, our approach differs from Li et al. in both the target of sparsification (edges vs. nodes) and the design philosophy (layer-specific adaptation vs. uniform treatment), resulting in a distinct theoretical and practical framework.
>
> We have condensed the above discussion and added to Section 3.2 after the bullet "layer-wise flexibility."

---

> ### Author Response · Authors · 2025-04-22
> **6. Clarification on Lemma 3.1 and Jumping Connections**
>
> ### Q: Can the authors clarify why Lemma 3.1 "reveals that the jumping connection allows for a more flexible sparsification strategy in deeper layers"? Generally, it's not evident to me why these results are only applicable to JK networks.
>
> We appreciate the reviewer's request for clarification. Lemma 3.1 indeed relies on the presence of a jumping connection structure, and the resulting conclusion—that the second layer can tolerate a higher sparsification level—is specific to such architectures.
>
> **In vanilla GCNs without skip connections, the output at the final layer depends compositionally on the outputs of all previous layers.** As a result, any sparsification error in the second layer is propagated through the transformation applied to the first layer, making it more difficult to isolate and control the impact of layer-specific pruning.
>
> **In contrast, the jumping connection allows the model output to aggregate features from both the first and second hidden layers**, rather than relying solely on the composition of the second layer over the first. This architectural decoupling ensures that the error introduced by sparsification at each layer can be bounded separately, as formalized in Equation (12). While the second layer still operates on the output of the first, the skip connection ensures that the first-layer representation is directly accessible at the output stage. This allows the model to tolerate larger approximation error (i.e., more pruning, higher $p_{ij}^2$) in the second layer without significantly affecting the total output error bound.
>
> Therefore, Lemma 3.1 does not hold for generic multi-layer GCNs without jump connections, and the increased flexibility in deeper-layer sparsification is a theoretical property unique to JK-style architectures.
>
> We have revised the statement of Lemma 3.1 and the discussion after that to enhance clarity.

---

> > ### Comment · Reviewer_gBfU · 2025-05-02
> >
> > Thank you for the comprehensive replies; I am convinced of the quality of this work. I have no further concerns.

---

### Decision · Action_Editor_upN4 · 2025-05-27

**Recommendation:** Accept as is

**Comment:**

This work provides a theoretical analysis of two-layer Graph Convolutional Networks (GCNs) with jumping knowledge connections and graph sparsification. It studies both the training dynamics and generalization performance of such networks. The theoretical results are solid and offer valuable insights into the role of jumping connections in determining sparsification requirements. Experimental results are also presented to support the theoretical findings. While the class of networks studied is somewhat limited, as noted by one reviewer, the theoretical analysis necessarily relies on certain assumptions. Nonetheless, the insights gained have the potential to inform broader classes of graph neural network architectures.

**Audience:**

Yes. The topic and results of the paper will be interesting to a substantial portion of audience of TMLR.

**Claims And Evidence:**

Yes, theoretical results are supported by formal proofs, and experimental results are provided to validate the claims presented in the paper.